# Global Update Tracking: A Decentralized Learning Algorithm for Heterogeneous Data

**Sai Aparna Aketi    Abolfazl Hashemi    Kaushik Roy**
Department of Electrical and Computer Engineering
Purdue University
West Lafayette, IN 47906
`{saketi, abolfazl, kaushik}@purdue.edu`

## Abstract

Decentralized learning enables training of deep learning models over large distributed datasets generated at different locations, without the need for a central server. However, in practical scenarios, the data distribution across these devices can be significantly different, leading to a degradation in model performance. In this paper, we focus on designing a decentralized learning algorithm that is less susceptible to variations in data distribution across devices. We propose Global Update Tracking (GUT), a novel tracking-based method that aims to mitigate the impact of heterogeneous data in decentralized learning without introducing any communication overhead. We demonstrate the effectiveness of the proposed technique through an exhaustive set of experiments on various Computer Vision datasets (CIFAR-10, CIFAR-100, Fashion MNIST, and ImageNette), model architectures, and network topologies. Our experiments show that the proposed method achieves state-of-the-art performance for decentralized learning on heterogeneous data via a $1 - 6\%$ improvement in test accuracy compared to other existing techniques.

## 1 Introduction

Decentralized learning is a branch of distributed optimization which focuses on learning from data distributed across multiple agents without a central server. It offers many advantages over the traditional centralized approach in core aspects such as data privacy, fault tolerance, and scalability [18]. It has been demonstrated that decentralized learning algorithms [15] can perform comparable to centralized algorithms on benchmark vision datasets. Decentralized Parallel Stochastic Gradient Descent (DSGD) presented in [15] combines SGD with a gossip averaging algorithm [26]. Further, the authors analytically show that the convergence rate of DSGD is similar to its centralized counterpart [5]. A momentum version of DSGD referred to as Decentralized Momentum Stochastic Gradient Descent (DSGDm) was proposed in [3]. The authors in [2] introduce Stochastic Gradient Push (SGP) which extends DSGD to directed and time-varying graphs. Recently, a unified framework for analyzing gossip-based decentralized SGD methods and the best-known convergence guarantees was presented in [11].

One of the key assumptions to achieve state-of-the-art performance by all the above-mentioned decentralized algorithms is that the data is independently and identically distributed (IID) across the agents. In particular, the data is assumed to be distributed in a uniform and random manner across the agents. This assumption does not hold in most real-world settings where the data distributions across the agents are significantly different (non-IID/heterogeneous) [9]. The effect of heterogeneous data in a peer-to-peer decentralized setup is a relatively under-studied problem and an active area of research.

Recently, there have been few efforts to bridge the performance gap between IID and non-IID data for a decentralized setup [16, 23, 19, 7, 1, 24]. Cross Gradient Aggregation [7] and Neighborhood

37th Conference on Neural Information Processing Systems (NeurIPS 2023).

Gradient Clustering [1] algorithms utilize the concept of cross-gradients to reduce the impact of heterogeneous data and show significant improvement in performance (test accuracy). However, these techniques incur $2\times$ communication cost than the standard decentralized algorithms such as DSGD. $D^2$ algorithm proposed in [23] is shown to be agnostic to data heterogeneity and can be employed in deep learning tasks. One of the major limitations of $D^2$ is that its convergence requires mixing topologies with negative eigenvalue bounded from below by $-\frac{1}{3}$. Additionally, it has been shown that $D^2$ performs worse than DSGD in some cases [16].

Tracking mechanisms such as Gradient Tracking (GT) [6, 19] and Momentum Tracking (MT) [22] have been proposed to tackle heterogeneous data in decentralized settings. But these algorithms improve the performance at the cost of $2\times$ communication overhead. The authors in [16] introduce Quasi-Global Momentum (QGM), a decentralized learning method that mimics the global synchronization of momentum buffer to mitigate the difficulties of decentralized learning on heterogeneous data. Recently, RelaySGD was presented in [24] that replaces the gossip averaging step with RelaySum. Since RelaySGD deals with the gossip averaging step, it is orthogonal to the aforementioned algorithms and can be used in synergy with them. QG-DSGDm [16] which incorporates QGM into DSGDm sets the current state-of-the-art for decentralized learning on heterogeneous data without increasing the communication cost. This work investigates the following question: *Can we improve decentralized learning on heterogeneous data through a tracking mechanism without any communication overhead?*

To that effect, we present *Global Update Tracking (GUT)*, a novel decentralized learning algorithm designed to improve performance under heterogeneous data distribution. Motivated by, yet distinct from, the gradient tracking mechanism, we propose to track the consensus model ($\bar{x}^t$) by tracking global/average model updates, where $x_i^t$ is the model parameters on agent $i$ at time step $t$ and $\bar{x}$ is the averaged model parameters. In the traditional tracking-based methods [19, 22] that track average gradients, each agent communicates both model parameters $x_i^t$ and the tracking variable $y_i^t$ with its neighbors resulting in $2\times$ communication overhead. The proposed *GUT* algorithm overcomes this bottleneck by allowing agents to store a copy of their neighbors' model parameters and then tracking the model updates instead of the gradients. This results in communicating only the tracking variable $y_i^t$ that yields the model update ($x_i^t - x_i^{t-1}$). We demonstrate the effectiveness of the proposed algorithm through an exhaustive set of experiments on various datasets, model architectures, and graph topologies. We also provide a detailed convergence analysis showing that the convergence rate of *GUT* algorithm is consistent with the state-of-the-art decentralized learning algorithms. Further, we show that *QG-GUTm* - Global Update Tracking with Quasi-Global momentum beats the current state-of-the-art decentralized learning algorithm (i.e., QG-DSGDm) on heterogeneous data under iso-communication cost.

## 1.1 Contributions

In summary, we make the following contributions.

- We propose *Global Update Tracking (GUT)*, a novel tracking-based decentralized learning algorithm to mitigate the impact of heterogeneous data distribution.

- We theoretically establish the non-asymptotic convergence rate of the proposed algorithm to a first-order solution.

- Through an exhaustive set of experiments on various datasets, model architectures, and graph topologies, we establish that the proposed Global Update Tracking with Quasi-Global momentum (*QG-GUTm*) outperforms the current state-of-the-art decentralized learning algorithm on a spectrum of heterogeneous data.

## 2 Background

In this section, we provide the background on the decentralized setup with peer-to-peer connections.

The main goal of decentralized machine learning is to learn a global model using the knowledge extracted from the locally stored data samples across $n$ agents while maintaining privacy constraints. In particular, we solve the optimization problem of minimizing the global loss function $f(x)$ distributed across $n$ agents as given in (1). Note that $F_i$ is a local loss function (for example, cross-entropy loss)

defined in terms of the data ($d_i$) sampled from the local dataset $D_i$ at agent $i$.

$$\min_{x \in \mathbb{R}^d} f(x) = \frac{1}{n} \sum_{i=1}^{n} f_i(x), \tag{1}$$

$$\text{where} \quad f_i(x) = \mathbb{E}_{d_i \sim D_i}[F_i(x; d_i)], \quad \text{for all } i.$$

The optimization problem is typically solved by combining stochastic gradient descent [4] with global consensus-based gossip averaging [26]. The communication topology is modeled as a graph $G = ([N], E)$ with edges $\{i, j\} \in E$ if and only if agents $i$ and $j$ are connected by a communication link exchanging the messages directly. We represent $\mathcal{N}(i)$ as the neighbors of agent $i$ including itself. It is assumed that the graph $G$ is strongly connected with self-loops i.e., there is a path from every agent to every other agent. The adjacency matrix of the graph $G$ is referred to as a mixing matrix $W$ where $w_{ij}$ is the weight associated with the edge $\{i, j\}$. Note that, weight 0 indicates the absence of a direct edge between the agents, and the elements of the Identity matrix are represented by $I_{ij}$. Similar to the majority of previous works in decentralized learning, the mixing matrix is assumed to be doubly stochastic. Further, the initial models and all the hyperparameters are synchronized at the beginning of the training. The communication among the agents is assumed to be synchronous.

Traditional decentralized algorithms such as DSGD [15] assume the data across the agents to be Independent and Identically Distributed (IID). In DSGD, each agent $i$ maintains local parameters $x_i^t \in \mathbb{R}^d$ and updates them as follows.

$$\text{DSGD:} \quad x_i^{t+1} = \sum_{j \in \mathcal{N}(i)} w_{ij}(x_j^t - \eta g_j^t); \quad g_i^t = \nabla F_j(x_i^t, d_i^t). \tag{2}$$

We focus on a decentralized setup with non-IID/heterogeneous data. In particular, the heterogeneity in the data distribution comes in the form of skewed label partition similar to [9]. Decentralized learning with the DSGD algorithm on heterogeneous data distribution results in performance degradation due to huge variations in the local gradients across the agents. To tackle this, authors in [16] propose a momentum-based optimization technique (QG-DSGDm) introducing Quasi-Global momentum as shown in (3).

$$\text{QG-DSGDm:} \quad x_i^{t+1} = \sum_{j \in \mathcal{N}(i)} w_{ij}[x_j^t - \eta(g_j^t + \beta m_j^{t-1})]; \quad m_i^t = \mu m_i^{t-1} + (1 - \mu)\frac{x_i^{t-1} - x_i^t}{\eta}. \tag{3}$$

QG-DSGDm improves the performance of decentralized learning on heterogeneous data without any communication overhead and is used as a baseline for comparison in this work.

Gradient Tracking (GT) mechanisms [19, 22] are also known to improve decentralized learning on heterogeneous data by reducing the variance between the local gradient and the averaged (global) gradient. To achieve this, the gradient tracking algorithm introduces a tracking variable $y_i^t$ that approximates the total gradient and is used to update the local parameters $x_i^t$ (refer to (4)).

$$\text{GT:} \quad x_i^{t+1} = \sum_{j \in \mathcal{N}(i)} w_{ij}(x_j^t - \eta y_j^t); \quad y_i^t = \sum_{j \in \mathcal{N}(i)} w_{ij} y_j^{t-1} - g_i^{t-1} + g_i^t. \tag{4}$$

The update rule of tracking variable is such that it recursively adds a correction term $\left(\sum_{j \in \mathcal{N}(i)} w_{ij} y_j^{t-1} - g_i^{t-1}\right)$ to the local gradient $g_i^t$, pushing $y_i^t$ to be closer to the global gradients ($\frac{1}{n} \sum_{j=1}^{n} g_j^t$). This requires each agent $i$ to communicate two sets of parameters $x_i^t$ and $y_i^t$ with its neighbors. Thus, the gradient tracking algorithm improves the decentralized learning on non-IID data at the cost of $2\times$ communication overhead.

## 3 Global Update Tracking

We present *Global Update Tracking (GUT)*, a novel algorithm for decentralized deep learning on non-IID data distribution. *GUT* is a communication-free tracking mechanism that aims to mitigate the difficulties of decentralized training when the data distributed across the agents is heterogeneous.

In order to attain the benefits of gradient tracking without communication overhead, we propose to apply the tracking mechanism with respect to the model updates $x_i^t - x_i^{t-1}$ instead of the gradients

---
**Algorithm 1** Global Update Tracking (*GUT*)
---
**Input:** Each agent $i \in [1, n]$ initializes model parameters $x_i^0$ and neighbors' copy $\hat{x}_j^0$, step size $\eta$, *GUT* scaling factor $\mu$, mixing matrix $W = [w_{ij}]_{i,j \in [1,n]}$, $\mathcal{N}(i)$ represents neighbors of $i$ including itself, and note $\hat{x}_i^t = x_i^t$.

Each agent simultaneously implements the TRAIN( ) procedure
1. **procedure** TRAIN( )
2.     **for** t = 0, 1, ..., T − 1 **do**
3.         $d_i^t \sim D_i$
4.         $g_i^t = \nabla_x F_i(\sum_{j \in \mathcal{N}(i)} w_{ij} \hat{x}_j^t; d_i^t)$
5.         $\delta_i^t = g_i^t - \frac{1}{\eta} \sum_{j \in \mathcal{N}(i)} w_{ij}(\hat{x}_j^t - x_i^t)$
6.         $y_i^t = \delta_i^t + \mu \Big[ \sum_{j \in \mathcal{N}(i)} w_{ij}(y_j^{t-1} - \frac{1}{\eta}(\hat{x}_j^t - x_i^t)) - \delta_i^{t-1} \Big]$
7.         SENDRECEIVE($y_i^t$)
8.         $x_i^{t+1} = x_i^t - \eta y_i^t$
9.         $\hat{x}_j^{t+1} = \hat{x}_j^t - \eta y_j^t \ \ \forall \ j \in \mathcal{N}(i) \backslash i$
10.    **end**
11. **return** $\frac{1}{n} \sum_{i=1}^n x_i^T$
---

$g_i^t$. Firstly, to design a tracking mechanism without additional communication cost, each agent $i$ communicates model updates instead of model parameters to its neighbors. An agent $i$ stores a copy of its neighbor's parameters as $\hat{x}_j$ and updates it using the received model updates to retrieve the current version of the neighbor's parameters as shown in line-9 of Algorithm 1. A memory-efficient implementation of the algorithm (Algorithm 4 in Appendix B) requires each agent to store $s_i = \sum_{j \in \mathcal{N}(i)} w_{ij} \hat{x}_j$ instead of storing each neighbor's copy separately requiring only $\mathcal{O}(1)$ additional memory [12].

Now, we define a variable $\delta_i^t$ on each agent $i$ that accumulates the local gradient update $g_i^t$ and the gossip averaging update $\sum_j (w_{ij} - I_{ij}) \hat{x}_j^t$ as shown in line-5 of Algorithm 1. Note that we can recover the DSGD update defined in (2) by using $\delta_i^t$ in the update rule i.e., $x_i^{t+1} = x_i^t - \eta \delta_i^t$. We then proceed to compute the tracking variable $y_i^t$, as described in line-6 of Algorithm 1, using the combined model update (local gradient part and gossip averaging part) reflected by $\delta_i^t$. The gossip averaging part of the update for each agent $i$ i.e., $\sum_j w_{ij}(\hat{x}_j^t - x_i^t)$ is computed with respect to its own model weights. To account for this in the computation of tracking variable $y_i^t$, the agents have to adjust the information received from the neighbors (i.e., $y_j^t$'s) to change the reference to itself. This is reflected as an additional term $\frac{1}{\eta}(\hat{x}_j^t - x_i^t)$ in the update rule given by line-6 of Algorithm 1. Further, we scale the correction term of the tracking variable by a factor $\mu$, a hyper-parameter, which is tuned to extract the maximum benefits of the proposed algorithm.

In summary, the update scheme of *GUT* can be re-formulated in the following matrix form where $X = [x_1, \ldots, x_n] \in \mathbb{R}^{d \times n}$ are the model parameters and $G = [g_1, \ldots, g_n] \in \mathbb{R}^{d \times n}$ are stochastic gradients.

$$X^{t+1} = X^t - \eta Y^t,$$
$$Y^{t+1} = G^{t+1} - \frac{1}{\eta}(W - I)X^{t+1} + \mu[WY^t - G^t - \frac{1}{\eta}(W - I)(X^{t+1} - X^t)]. \tag{5}$$

Finally, we show that integrating the proposed *GUT* algorithm with Quasi-Global Momentum improves the current state-of-the-art significantly without any communication overhead. The pseudo-code for the momentum version of our algorithm (*QG-GUTm*) is presented in Appendix B.

## 4 Convergence Guarantees

This section provides the convergence analysis for the proposed *GUT* Algorithm. We assume that the following standard assumptions hold:

**Assumption 1** (Lipschitz Gradients). *Each function $f_i(x)$ is L-smooth i.e., $||\nabla f_i(y) - \nabla f_i(x)|| \leq L||y - x||$.*

**Assumption 2** (Bounded Variance). *The stochastic gradients are unbiased and their variance is assumed to be bounded.*

$$\mathbb{E}_{d\sim D_i}||\nabla F_i(x; d) - \nabla f_i(x)||^2 \leq \sigma^2 \quad \forall i \in [1, n], \tag{6}$$

$$\frac{1}{n}\sum_{i=1}^{n}||\nabla f_i(x) - \nabla f(x)||^2 \leq \zeta^2. \tag{7}$$

**Assumption 3** (Doubly Stochastic Mixing Matrix). *The mixing matrix $W$ is a real doubly stochastic matrix with $\lambda_1(W) = 1$ and*

$$\max\{|\lambda_2(W)|, |\lambda_n(W)|\} \leq 1 - \rho < 1, \tag{8}$$

*where $\lambda_i(W)$ is the $i^{th}$ largest eigenvalue of W and $\rho$ is the spectral gap.*

The above assumptions are commonly used in most decentralized learning setups. Theorem 1 presents the convergence of the proposed *GUT* algorithm and the proof is detailed in Appendix A.

**Theorem 1.** *(Convergence of GUT algorithm) Given Assumptions 1, 2, and 3 let step size $\eta \leq \frac{\rho}{7L}$ and the scaling factor $\frac{\mu}{1-\mu} \leq \frac{\rho}{42}$. For all $T \geq 1$, we have*

$$\frac{1}{T}\sum_{t=0}^{T-1}\mathbb{E}||\nabla f(\bar{x}^t)||^2 \leq \frac{4}{\eta T}(f(\bar{x}^0) - f^*) + \eta\frac{4L\sigma^2}{n} + \eta^2\frac{1248L^2}{\rho^2}(\zeta^2 + \sigma^2(2 - \mu)), \tag{9}$$

*where $f(\bar{x}^0) - f^*$ is the sub-optimality gap, $\bar{x}$ is the average/consensus model parameters.*

The result of the Theorem 1 shows that the averaged gradient of the averaged model is upper-bounded by the sub-optimality gap (the difference between the initial objective function value and the optimal value), the sampling variance ($\sigma$), and gradient variations across the agents representing data heterogeneity ($\zeta$). Further, we present a corollary to show the convergence rate of *GUT* in terms of the number of iterations.

**Corollary 1.** *Suppose that the step size satisfies $\eta = \mathcal{O}\left(\sqrt{\frac{n}{T}}\right)$ For a sufficiently large T we have,*

$$\frac{1}{T}\sum_{t=0}^{T-1}\mathbb{E}||\nabla f(\bar{x}^t)||^2 \leq \mathcal{O}\left(\frac{1}{\sqrt{nT}} + \frac{1}{T}\right). \tag{10}$$

Corollary 1 indicates that the *GUT* algorithm achieves linear speedup with a convergence rate of $\mathcal{O}(\frac{1}{\sqrt{nT}})$ when $T$ is sufficiently large and is independent of communication topology. In other words, the communication complexity to find an $\epsilon$-first order solution, i.e., $\mathbb{E}||\nabla f(\bar{x})||^2 \leq \epsilon$ is $\mathcal{O}(\frac{\sigma^2}{n\epsilon^2})$. This convergence rate is similar to the well-known best result for decentralized SGD algorithms [15] in the literature.

## 5 Experiments

In this section, we analyze the performance of the proposed *GUT* and *QG-GUTm* techniques and compare them with the baseline DSGD algorithm [15] and the current state-of-the-art QG-DSGDm [16] respectively. The source code is available at `https://github.com/aparna-aketi/global_update_tracking`

### 5.1 Experimental Setup

The efficiency of the proposed method is demonstrated through our experiments on a diverse set of datasets, model architectures, graph topologies, and graph sizes. We present the analysis on – (a) Datasets: CIFAR-10, CIFAR-100, Fashion MNIST, and Imagenette. (b) Model architectures: VGG-11, ResNet-20, LeNet-5 and, MobileNet-V2. All the models use Evonorm [17] as the activation-normalization layer as it is shown to be better suited for decentralized learning on heterogeneous data.

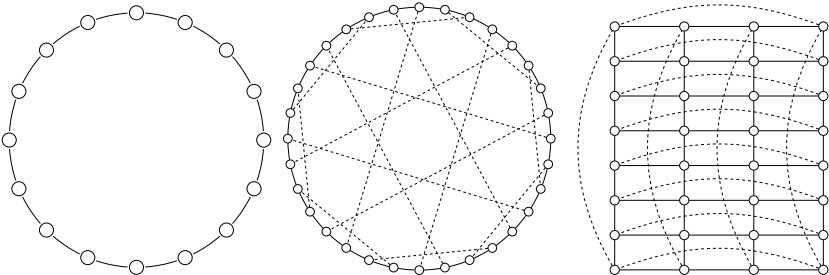

Figure 1: Ring Graph (left), Dyck Graph (center), and Torus Graph (right).

(c) Graph topologies: Ring graph with 2 peers per agent, Dyck graph with 3 peers per agent, and Torus graph with 4 peers per agent (refer Figure 1). (d) Number of agents: 16-40 agents. We use the Dirichlet distribution to generate disjoint non-IID data across the agents. The created data partition across the agents is fixed, non-overlapping, and never shuffled across agents during the training. The degree of heterogeneity is regulated by the value of $\alpha$ – the smaller the $\alpha$ the larger the non-IIDness across the agents. We report the test accuracy of the consensus model averaged over three randomly chosen seeds. The details of the decentralized setup and hyperparameters for all the experiments are presented in Appendix C.

## 5.2 Average Consensus Task

We first consider an average consensus task that is isolated from the learning through stochastic gradient descent. Here the aim is that all the agents should reach a consensus which is the average value of the initial information each agent holds. The following equations show the simplified version of *GUT* (11) and *QG-GUTm* (12) after removing the gradient update part.

$$X^{t+1} = X^t + Y^t; \quad Y^t = (W - I)X^t + \mu[WY^{t-1} - (W - I)(X^{t-1} - X^t)], \tag{11}$$

$$X^{t+1} = X^t + \hat{M}^t; \quad M^t = \beta M^{t-1} + (1 - \beta)(X^t - X^{t-1})$$

$$\hat{M}^t = \beta M^t + (1 - \beta)[(W - I)X^t + \mu(W\hat{M}^{t-1} - (W - I)(X^{t-1} - X^t))]. \tag{12}$$

Note that setting the hyper-parameter $\mu$ as 0 in the (11), 12 gives simple gossip[26] and quasi-global gossip [16] respectively and all the agents communicate $X^t - X^{t-1}$ at iteration $t$ with their neighbors.

Figure.2 shows the average consensus error i.e., $\frac{1}{n}||X^t - \bar{X}||_F^2$ over time for the average consensus task on the Ring topology with respect to various algorithms. We observe that the gossip averaging with *GUT* converges faster than simple gossip averaging. Figure.2(c) illustrates that for graphs with a smaller spectral gap (which corresponds to more agents), the proposed *QG-GUTm* can converge faster than quasi-global gossip (gossip with QGM) resulting in better decentralized optimization.

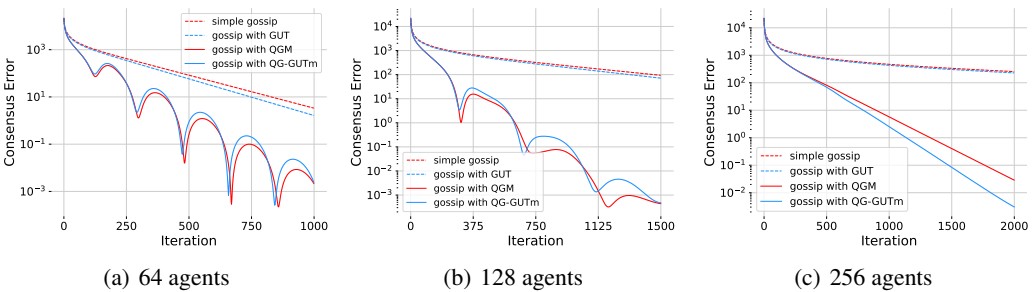

(a) 64 agents      (b) 128 agents      (c) 256 agents

Figure 2: Decentralized average consensus problem on an undirected ring topology

## 5.3 Decentralized Deep Learning Results

We evaluate the efficiency of *GUT* and its quasi-global momentum version *QG-GUTm* with the help of an exhaustive set of experiments. We compare *GUT* with DSGD and the momentum version *QG-GUTm* with QG-DSGDm to show that the proposed method outperforms the current state-of-the-art.

Table. 1 shows the average test accuracy for training ResNet-20 and VGG-11 models on the CIFAR-10 dataset with varying degrees of non-IIDness over ring topology of 16 and 32 agents. We observe that *GUT* consistently outperforms DSGD for all models, graph sizes, and degree of heterogeneity with a significant performance gain varying from $1 - 18\%$. The quasi-global momentum version of our algorithm, *QG-GUTm*, beats QG-DSGDm with $1 - 3.5\%$ improvement in the case of the CIFAR-10 dataset partitioned with a higher degree of heterogeneity ($\alpha = 0.1, 0.01$).

Table 1: Test accuracy of different decentralized algorithms evaluated on CIFAR-10, distributed with different degrees of heterogeneity (non-IID) for various models over ring topologies. The results are averaged over three seeds where the standard deviation is indicated. We also include the results of the IID baseline as DSGDm (IID) where the local data is randomly partitioned independent of $\alpha$.

| Agents ($n$) | Method | ResNet-20 | | |
| --- | --- | --- | --- | --- |
| | | $\alpha = 1$ | $\alpha = 0.1$ | $\alpha = 0.01$ |
| 16 | DSGDm (IID) | | $89.75 \pm 0.29$ | |
| | DSGD [15] | $84.17 \pm 0.32$ | $72.21 \pm 2.37$ | $54.66 \pm 4.74$ |
| | *GUT (ours)* | $84.72 \pm 0.20$ | $81.86 \pm 1.99$ | $70.16 \pm 4.94$ |
| | QG-DSGDm [16] | $\mathbf{88.23} \pm 0.51$ | $84.21 \pm 2.12$ | $79.85 \pm 2.11$ |
| | *QG-GUTm (ours)* | $88.22 \pm 0.36$ | $\mathbf{86.44} \pm 0.36$ | $\mathbf{81.04} \pm 1.66$ |
| 32 | DSGDm (IID) | | $88.52 \pm 0.23$ | |
| | DSGD [15] | $78.25 \pm 0.42$ | $62.97 \pm 1.90$ | $42.58 \pm 1.84$ |
| | *GUT (ours)* | $79.24 \pm 0.33$ | $76.07 \pm 0.23$ | $60.72 \pm 1.03$ |
| | QG-DSGDm [16] | $87.15 \pm 0.33$ | $83.50 \pm 1.04$ | $69.99 \pm 0.60$ |
| | *QG-GUTm (ours)* | $\mathbf{87.48} \pm 0.33$ | $\mathbf{84.94} \pm 0.60$ | $\mathbf{72.04} \pm 3.18$ |
| Agents ($n$) | Method | VGG-11 | | |
| | | $\alpha = 1$ | $\alpha = 0.1$ | $\alpha = 0.01$ |
| 16 | DSGDm (IID) | | $85.76 \pm 0.28$ | |
| | DSGD [15] | $81.78 \pm 0.29$ | $76.20 \pm 0.81$ | $68.93 \pm 1.23$ |
| | *GUT (ours)* | $82.12 \pm 0.09$ | $81.24 \pm 0.95$ | $76.62 \pm 1.37$ |
| | QG-DSGDm [16] | $84.23 \pm 0.47$ | $81.70 \pm 0.79$ | $77.08 \pm 3.19$ |
| | *QG-GUTm (ours)* | $\mathbf{84.46} \pm 0.33$ | $\mathbf{83.05} \pm 0.48$ | $\mathbf{78.32} \pm 1.03$ |
| 32 | DSGDm (IID) | | $84.75 \pm 0.30$ | |
| | DSGD [15] | $79.75 \pm 0.56$ | $73.37 \pm 1.02$ | $59.93 \pm 1.60$ |
| | *GUT (ours)* | $80.37 \pm 0.33$ | $79.55 \pm 1.00$ | $73.59 \pm 1.26$ |
| | QG-DSGDm [16] | $83.67 \pm 0.28$ | $80.82 \pm 0.19$ | $74.25 \pm 2.02$ |
| | *QG-GUTm (ours)* | $\mathbf{84.32} \pm 0.11$ | $\mathbf{83.39} \pm 0.38$ | $\mathbf{77.41} \pm 3.44$ |

Table 2: Average test accuracy of different decentralized algorithms evaluated on CIFAR-10 dataset trained on ResNet-20 over various graph topologies

| Method | Dyck Graph (32 agents) | | Torus (32 agents) | |
| --- | --- | --- | --- | --- |
| | $\alpha = 0.1$ | $\alpha = 0.01$ | $\alpha = 0.1$ | $\alpha = 0.01$ |
| DSGDm [15] | $77.31 \pm 1.72$ | $51.27 \pm 2.51$ | $75.37 \pm 3.11$ | $49.32 \pm 2.88$ |
| QG-DSGDm [16] | $86.49 \pm 0.81$ | $81.32 \pm 1.50$ | $86.88 \pm 0.30$ | $85.20 \pm 0.56$ |
| *QG-GUTm (ours)* | $\mathbf{86.93} \pm 0.53$ | $\mathbf{84.80} \pm 0.47$ | $\mathbf{87.75} \pm 0.42$ | $\mathbf{86.20} \pm 0.82$ |

Table 3: Average test accuracy of different decentralized algorithms evaluated on various datasets, distributed with different degrees of heterogeneity over 16 agents ring topology

| Method | Fashion MNIST (LeNet-5) | | CIFAR-100 (ResNet-20) | | Imagenette (MobileNet-V2) | |
| --- | --- | --- | --- | --- | --- | --- |
| | $\alpha = 0.1$ | $\alpha = 0.01$ | $\alpha = 0.1$ | $\alpha = 0.01$ | $\alpha = 0.1$ | $\alpha = 0.01$ |
| DSGDm [15] | $86.59 \pm 0.92$ | $77.00 \pm 3.53$ | $47.93 \pm 1.69$ | $42.56 \pm 2.71$ | $66.02 \pm 4.59$ | $38.69 \pm 11.8$ |
| QG-DSGDm [16] | $89.94 \pm 0.44$ | $83.43 \pm 0.94$ | $53.19 \pm 1.68$ | $44.17 \pm 3.64$ | $63.60 \pm 4.50$ | $39.49 \pm 4.57$ |
| *QG-GUTm* | $\mathbf{90.11} \pm 0.02$ | $\mathbf{84.60} \pm 1.00$ | $\mathbf{53.40} \pm 1.23$ | $\mathbf{50.45} \pm 1.30$ | $\mathbf{66.52} \pm 3.68$ | $\mathbf{43.85} \pm 8.24$ |

We present the experimental results on various graph topologies and datasets to demonstrate the scalability and generalizability of *QG-GUTm*. We train the CIFAR-10 dataset on ResNet-20 over the Dyck graph and Torus graph to exemplify the impact of connectivity on the proposed technique. As shown in Table. 2, we obtain $0.5 - 3.5\%$ performance gains with varying connectivity (or spectral gap).

Further, we evaluate *QG-GUTm* on various image datasets such as Fashion MNIST, and Imagenette and on challenging datasets such as CIFAR-100. Table. 3 shows that *QG-GUTm* outperforms QG-DSGDm by $0.2 - 6.2\%$ across various datasets. Therefore, in a decentralized deep learning setup, the proposed *GUT* and *QG-GUTm* algorithms are more robust to heterogeneity in the data distribution and can outperform all the comparison methods with an average improvement of $2\%$.

## 5.4 Ablation Study

First, we analyze different ways of utilizing or tracking model update information as shown in Table. 4. We present two different update rules apart from *GUT* and *DSGD*[15] and also compare them with gradient tracking [19]. Rule-a applies the proposed tracking mechanism on model updates but does not change the reference of tracking variable $y_j$ received from the neighbors to itself (refer sec. 3 for details on changing the reference). In the case of Rule-b, each agent computes the difference between the averaged neighborhood model update ($W(X^t - X^{t-1})$) along with its own model update ($X^t - X^{t-1}$) and adds the difference between the two as a bias correction. Table. 4 shows that such naive ways of tracking or bias correction update rules (rule-a,b) do not improve the performance of decentralized learning on heterogeneous data. This confirms our findings that the *GUT* technique is an effective and provable way to track the consensus model and can outperform the gradient tracking mechanism without any communication overhead.

Table 4: Analyzing the different variations of model updates. Evaluating test accuracy on CIFAR-10 dataset trained on ResNet-20 over a 16 agent ring topology with $\alpha = 0.1$

| Method | Update Rule | Communication Parameters (Cost) | Test accuracy |
|---|---|---|---|
| DSGD | $X^{t+1} = X^t - \eta Y^t$ 
 $Y^t = G^t - \frac{1}{\eta}(W-I)X^t$ | $X^t$ (1×) | $72.21 \pm 2.37$ |
| Rule-a | $X^{t+1} = X^t - \eta Y^t$ 
 $Y^t = G^t - \frac{1}{\eta}(W-I)X^t + \mu[WY^{t-1} - (G^{t-1} - \frac{1}{\eta}(W-I)X^{t-1})]$ | $Y^t$ (1×) | $72.78 \pm 0.80$ |
| Rule-b | $X^{t+1} = X^t - \eta Y^t$ 
 $Y^t = G^t - \frac{1}{\eta}(W-I)X^t + \mu[-\frac{1}{\eta}(W-I)(X^t - X^{t-1})]$ | $Y^t$ (1×) | $72.62 \pm 1.16$ |
| GUT | $X^{t+1} = X^t - \eta Y^t$ 
 $Y^t = G^t - \frac{1}{\eta}(W-I)X^t + \mu[WY^{t-1} - G^{t-1} - \frac{1}{\eta}(W-I)(X^t - X^{t-1})]$ | $Y^t$ (1×) | $\mathbf{81.86 \pm 1.99}$ |
| Gradient Tracking | $X^{t+1} = X^t - \eta[Y^t - \frac{1}{\eta}(W-I)X^t]$ 
 $Y^t = G^t + WY^{t-1} - G^{t-1}$ | $X^t, Y^t$ (2×) | $80.61 \pm 2.41$ |

Table 5: Evaluating Global Update Tracking (GUT) with various versions of momentum using CIFAR-10 dataset trained on ResNet-20 architecture over 16 agents ring topology

| Method | Local Momentum | Nesterov | Quasi-Global Momentum | Global Update Tracking | Test Accuracy $\alpha = 0.1$ |
|---|---|---|---|---|---|
| DSGD | x | x | x | x | $72.21 \pm 2.37$ |
| DSGDm | ✓ | x | x | x | $79.87 \pm 1.73$ |
| DSGDm-N | ✓ | ✓ | x | x | $81.31 \pm 0.51$ |
| QG-DSGDm | x | x | ✓ | x | $84.21 \pm 2.12$ |
| QG-DSGDm-N | x | ✓ | ✓ | x | $85.12 \pm 1.11$ |
| GUT | x | x | x | ✓ | $81.86 \pm 1.99$ |
| GUTm | ✓ | x | x | ✓ | $79.95 \pm 1.67$ |
| GUTm-N | ✓ | ✓ | x | ✓ | $82.08 \pm 1.74$ |
| QG-GUTm | x | x | ✓ | ✓ | $86.44 \pm 0.36$ |
| QG-GUTm-N | x | ✓ | ✓ | ✓ | $\mathbf{86.55 \pm 0.49}$ |

We then proceed to investigate the effect of different variants of momentum with *GUT*. From Table. 5 (refer to Appendix D for more results), we can conclude that the quasi-global variant of Global Update Tracking always surpasses the other methods. This indicates that the proposed *GUT* algorithm accelerates decentralized optimization and can be used in synergy with quasi-global momentum to achieve maximal performance gains.

Furthermore, Figure 3(a) illustrates the effect of scaling $\mu$ on the test accuracy with *QG-GUTm* and note that $\mu = 0$ shows the test accuracy for QG-DSGDm. Figure 3(b), 3(c) showcase the scalability

of *QG-GUTm* on different graph sizes and model sizes. *QG-GUTm* outperforms QG-DSGDm by
$\sim 1.7\%$ over different graph sizes and $\sim 1.4\%$ over different model sizes.

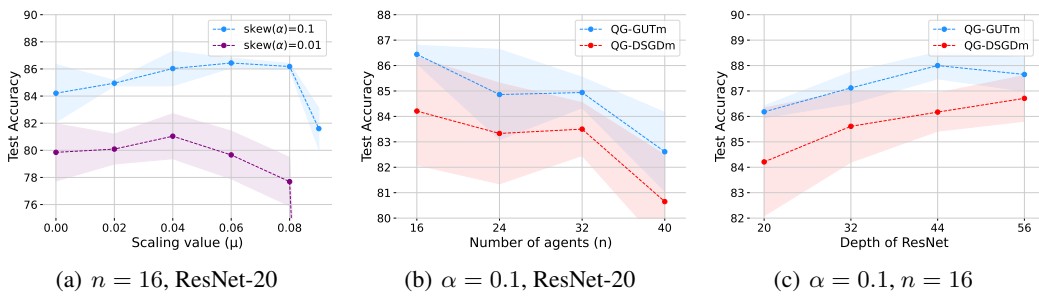

(a) $n = 16$, ResNet-20      (b) $\alpha = 0.1$, ResNet-20      (c) $\alpha = 0.1$, $n = 16$

Figure 3: Ablation study on the hyper-parameter $\mu$, number of agents $n$ and model size. The test
accuracy is reported for the CIFAR-10 dataset trained on ResNet architecture over ring topology.

## 6 Discussion and Limitations

We demonstrated the superiority of the Global Update Tracking (*GUT*) algorithm through an elaborate
set of experiments and ablation studies. In our experiments, we focused on doubly-stochastic and
symmetric graph structures. The proposed *GUT* algorithm can be easily extended to directed and
time-varying graphs by combining it with stochastic gradient push (SGP) [2]. Further, the additional
terms added by *GUT* can also be interpreted as a bias correction mechanism where the added bias
pushes the local model towards the consensus (averaged) model. The matrix representation of this
interpretation of *GUT* is given by (13). and analyzed in Lemma 1.

$$X^{t+1} = WX^t - \eta(G^t + \mu B^t); \quad B^{t+1} = -\frac{1}{\eta}[(2W - I)(X^{t+1} - X^t) + \eta G^t]. \quad (13)$$

**Lemma 1.** *Given assumptions 3, we define* $\bar{b}^t = B^t \frac{1}{n} \mathbb{1}\mathbb{1}^T$, *where* $\mathbb{1}$ *is a vector of all ones. For all t,*
*we have:* $\bar{b}^t = \mu \bar{b}^{t-1}$.

A complete proof for Lemma 1 can be found in Appendix A.2. Lemma 1 highlights that the average
bias added by *GUT* is zero as $\bar{b}^0$ is zero. Hence, the *GUT* algorithm crucially preserves the average
value of the decentralized system. A feature we leverage to establish Theorem 1.

There are two potential limitations of the *GUT* algorithm - a) memory overhead and b) introduction
of an additional hyper-parameter. *GUT* requires the agents to keep a copy of the averaged model
parameters of their neighbors which adds an extra memory buffer of the size of model parameters.
The storage of the tracking variable also adds to the memory overhead, requiring additional memory
equivalent to the size of model parameters. We also introduce a new hyper-parameter $\mu$ which has to be
tuned similarly to the learning rate or momentum coefficient tuning. Besides, the theoretical analysis
presented for the *GUT* algorithm does not consider momentum and assumes the communication to be
synchronous. We leave the theoretical analysis of *QG-GUTm* and formulation of the asynchronous
version of *GUT* as a future research direction.

## 7 Conclusion

Decentralized learning on heterogeneous data is the key to launching ML training on edge devices
and thereby efficiently leveraging the humongous amounts of user-generated private data. In this
paper, we propose *Global Update Tracking* (*GUT*), a novel decentralized algorithm designed to
improve learning over heterogeneous data distributions. The convergence analysis presented in the
paper shows that the proposed algorithm matches the best-known rate for decentralized algorithms.
Additionally, the paper introduces a quasi-global momentum version of the algorithm, QG-GUTm, to
further enhance the performance gains. The empirical evidence from experiments on different model
architectures, datasets, and topologies demonstrates the superior performance of both algorithms.
In summary, the proposed algorithm and its quasi-global momentum version have the potential to
facilitate more scalable and efficient decentralized learning on edge devices.

## Acknowledgements

This work was supported in part by, the Center for the Co-Design of Cognitive Systems (COCOSYS), a DARPA-sponsored JUMP center, the Semiconductor Research Corporation (SRC), and the National Science Foundation.

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

# A Convergence Rate Proof

In this work, we solve the optimization problem of minimizing global loss function $f(x)$ distributed across $n$ agents as given below. Note that $F_i$ is a local loss function (for example, cross-entropy loss) defined in terms of the data sampled $(d_i)$ from the local dataset $D_i$ at agent $i$.

$$\min_{x \in \mathbb{R}^d} f(x) = \frac{1}{n} \sum_{i=1}^{n} f_i(x),$$

$$and \ \ f_i(x) = \mathbb{E}_{d_i \in D_i}[F_i(x; d_i)] \ \ \forall i.$$

We reiterate the update scheme of *GUT* presented in Algorithm. 1 in a matrix form:

$$X^{t+1} = X^t - \eta Y^t$$

$$Y^t = \Delta^t + \mu[WY^{t-1} - \frac{1}{\eta}(W - I)X^t - \Delta^{t-1}] \tag{14}$$

$$\Delta^t = G^t - \frac{1}{\eta}(W - I)X^t,$$

where $W$ is the mixing matrix, $I$ is the identity matrix, $X = [x_1, x_2, \ldots, x_n] \in \mathbb{R}^{d \times n}$ is the matrix containing model parameters, $x_i \in \mathbb{R}^d$ is model parameters of agent $i$, $Y = [y_1, y_2, \ldots, y_n] \in \mathbb{R}^{d \times n}$ is the matrix containing tracking variables, $G = [g_1, g_2, \ldots, g_n] \in \mathbb{R}^{d \times n}$ is the matrix containing local gradients, $\mu$ is the *GUT* scaling factor, $\eta$ is the learning rate. Now, we rewrite the above equation in the form of a bias correction update,

$$X^{t+1} = WX^t - \eta(G^t + \mu B^t)$$

$$B^t = -\frac{1}{\eta}[(2W - I)(X^t - X^{t-1}) + \eta G^{t-1}]. \tag{15}$$

## A.1 Assumptions

We assume that the following statements hold:

**Assumption 1 - Lipschitz Gradients:** Each function $f_i(x)$ is L-smooth i.e., $||\nabla f_i(y) - \nabla f_i(x)|| \leq L||y - x||$. Equivalently,

$$f_i(y) \leq f_i(x) + \langle \nabla f_i(x), y - x \rangle + \frac{L}{2}||y - x||^2 \tag{16}$$

**Assumption 2 - Bounded Variance:** The variance of the stochastic gradients is assumed to be bounded.

$$\mathbb{E}_{d \sim D_i}||\nabla F_i(x; d) - \nabla f_i(x)||^2 \leq \sigma^2 \ \ \forall i \in [1, n]$$

$$\frac{1}{n} \sum_{i=1}^{n} ||\nabla f_i(x) - \nabla f(x)||^2 \leq \zeta^2$$

**Assumption 3 - Doubly Stochastic Mixing Matrix:** The mixing matrix $W$ is a real doubly stochastic matrix with $\lambda_1(W) = 1$ and

$$max\{|\lambda_2(W)|, |\lambda_n(W)|\} \leq 1 - \rho < 1$$

where $\lambda_i(W)$ is the $i^{th}$ largest eigenvalue of W and $\rho$ is the spectral gap. The mixing matrix satisfies $\mathbb{E}_W ||ZW - \bar{Z}||_F^2 \leq (1 - \rho)||ZW - \bar{Z}||_F^2$, where $\bar{Z} = Z\frac{1}{n}\mathbb{1}\mathbb{1}^T$. We also have $W\mathbb{1} = \mathbb{1}$ and $W^T\mathbb{1} = \mathbb{1}$.

Further, we define the average gradients $\bar{g}^t = \frac{1}{n} \sum_{i=1}^{n} \nabla F_i(x_i^t, d_i^t)$ where $d_i^t$ is sampled mini-batch of data on node $i$

## A.2 Proof of Lemma 1

**Lemma 1:** *Given assumptions 3, we define* $\bar{b}^t = B^t \frac{1}{n} \mathbb{1}\mathbb{1}^T$, *where* $\mathbb{1}$ *is a vector of all ones. For all t, we have:* $\bar{b}^t = \mu \bar{b}^{t-1}$.

*Proof.* Starting from the definition of $B^t$

$$B^t = -\frac{1}{\eta}[(2W - I)(X^t - X^{t-1}) + \eta G^{t-1}]$$

multiply $\frac{1}{n}\mathbb{1}\mathbb{1}^T$ on both sides

$$\bar{b}^t = -\frac{1}{\eta}[\bar{x}^t - \bar{x}^{t-1} + \eta\bar{g}^{t-1}] \quad (\because (2W - I)\mathbb{1} = \mathbb{1})$$

now, multiplying $\frac{1}{n}\mathbb{1}\mathbb{1}^T$ to $X^{t+1} = WX^t - \eta(G^t + \mu B^t)$

$$\bar{x}^{t+1} = \bar{x}^t - \eta\bar{g}^t - \eta\mu\bar{b}^t \implies \bar{x}^t - \bar{x}^{t-1} + \eta\bar{g}^{t-1} = -\eta\mu\bar{b}^{t-1}$$

$$\implies \bar{b}^t = \mu\bar{b}^{t-1}$$

$\square$

Given that $\bar{b}^0 = 0$, the average bias is zero at each iteration. This indicates that the proposed algorithm *GUT* preserves the average of the system in an average consensus task.

## A.3 Proof of Theorem 1

This section presents the detailed proof of the convergence bounds of *GUT* algorithm given by Theorem 1. Firstly, we analyze the one-step progress of the averaged model parameters $\bar{x}$. Note that, $\bar{X}^t = [\bar{x}^t, \bar{x}^t, \ldots, \bar{x}^t] \in \mathbb{R}^{d \times n}$ and $\bar{x}^t = \frac{1}{n}\sum_{i=1}^{n} x_i^t$

**Lemma 2.** *Given assumptions 1-3 and* $\eta \leq \frac{1}{4L}$, *we have*

$\mathbb{E}f(\bar{x}^{t+1}) \leq \mathbb{E}f(\bar{x}^t) - \frac{\eta}{4}\mathbb{E}||\nabla f(\bar{x}^t)||^2 - \frac{\eta}{4}\mathbb{E}||\frac{1}{n}\sum_{i=1}^{n}\nabla f(x_i^t)||^2 + \frac{L\eta^2\sigma^2}{n} + \frac{3L\eta^2}{n}||X^t - \bar{X}^t||_F^2$.

*Proof.* From the definition of $X^{t+1}$, we have

$$X^{t+1} = WX^t - \eta[G^t + \mu B^t]$$

$$\implies \bar{x}^{t+1} = \bar{x}^t - \eta\bar{g}^t \quad (\because \bar{b}^t = 0 \text{ from Lemma 1})$$

using L-smoothness assumption given by (16)

$$\mathbb{E}f(\bar{x}^{t+1}) \leq \mathbb{E}f(\bar{x}^t) + \mathbb{E}\langle\nabla f(\bar{x}^t), \bar{x}^{t+1} - \bar{x}^t\rangle + \frac{L}{2}\mathbb{E}||\bar{x}^{t+1} - \bar{x}^t||^2$$

$$= \mathbb{E}f(\bar{x}^t) + \mathbb{E}\langle\nabla f(\bar{x}^t), -\eta\bar{g}^t\rangle + \frac{L\eta^2}{2}\mathbb{E}||\bar{g}^t||^2$$

$$= \mathbb{E}f(\bar{x}^t) - \eta\mathbb{E}\langle\nabla f(\bar{x}^t), \mathbb{E}[\bar{g}^t]\rangle + \frac{L\eta^2}{2}\mathbb{E}||\frac{1}{n}\sum_{i=1}^{n}\nabla F_i(x_i^t)||^2$$

$$= \mathbb{E}f(\bar{x}^t) - \eta\frac{1}{n}\sum_{i=1}^{n}\mathbb{E}\langle\nabla f(\bar{x}^t), \nabla f_i(x_i^t)\rangle + \frac{L\eta^2}{2}\mathbb{E}||\frac{1}{n}\sum_{i=1}^{n}(\nabla F_i(x_i^t) \pm \nabla f_i(x_i^t))||^2$$

$$\overset{(a)}{\leq} \mathbb{E}f(\bar{x}^t) - \eta\mathbb{E}\langle\nabla f(\bar{x}^t), \frac{1}{n}\sum_{i=1}^{n}\nabla f_i(x_i^t)\rangle + \frac{L\eta^2}{2}\mathbb{E}||\frac{1}{n}\sum_{i=1}^{n}\nabla f_i(x_i^t)||^2 + \frac{L\eta^2\sigma^2}{n}$$

$$\overset{(b)}{=} \mathbb{E}f(\bar{x}^t) + \frac{L\eta^2\sigma^2}{n} + \frac{L\eta^2}{2}\mathbb{E}||\frac{1}{n}\sum_{i=1}^{n}\nabla f_i(x_i^t)||^2 - \frac{\eta}{2}\mathbb{E}||\nabla f(\bar{x}^t)||^2$$

$$- \frac{\eta}{2}\mathbb{E}||\frac{1}{n}\sum_{i=1}^{n}\nabla f_i(x_i^t)||^2 + \frac{\eta}{2}\mathbb{E}||\frac{1}{n}\sum_{i=1}^{n}(\nabla f_i(x_i^t) - \nabla f(\bar{x}^t))||^2$$

$$\overset{(c)}{\leq} \mathbb{E}f(\bar{x}^t) + \frac{L\eta^2\sigma^2}{n} + (\frac{L\eta^2}{2} - \frac{\eta}{2})\mathbb{E}||\frac{1}{n}\sum_{i=1}^{n}\nabla f_i(x_i^t)||^2 - \frac{\eta}{2}\mathbb{E}||\nabla f(\bar{x}^t)||^2$$

$$+ \frac{\eta}{2n}\sum_{i=1}^{n}\mathbb{E}||\nabla f_i(x_i^t) - \nabla f(\bar{x}^t)||^2$$

$$\overset{(d)}{\leq} \mathbb{E}f(\bar{x}^t) + \frac{L\eta^2\sigma^2}{n} + (\frac{L\eta^2}{2} - \frac{\eta}{2})\mathbb{E}||\frac{1}{n}\sum_{i=1}^{n}\nabla f_i(x_i^t)||^2 - \frac{\eta}{2}\mathbb{E}||\nabla f(\bar{x}^t)||^2$$

$$+ \frac{L^2\eta}{2n}\sum_{i=1}^{n}\mathbb{E}||x_i^t - \bar{x}^t||^2$$

$$\overset{(e)}{\leq} \mathbb{E}f(\bar{x}^t) - \frac{\eta}{4}\mathbb{E}||\frac{1}{n}\sum_{i=1}^{n}\nabla f_i(x_i^t)||^2 - \frac{\eta}{4}\mathbb{E}||\nabla f(\bar{x}^t)||^2 + \frac{L\eta^2\sigma^2}{n}$$

$$+ \frac{3L^2\eta}{n}\sum_{i=1}^{n}\mathbb{E}||X^t - \bar{X}^t||_F^2$$

(a) uses assumption-2 ((6)). (b) uses the fact that $-2\langle a, b \rangle = -||a||^2 - ||b||^2 + ||a - b||^2$. (c) uses Jensen's inequality. (d) uses L-smoothness condition. (e) follows from the assumption that $\eta \leq \frac{1}{4L}$. $\qquad\square$

Now, we proceed to bound the consensus error through Lemma 3.

**Lemma 3.** *Given assumptions 1-3 and $\eta \leq \frac{\rho}{7L}$, we have*
$$\frac{1}{n}\mathbb{E}||X^{t+1} - \bar{X}^{t+1}||_F^2 \leq \frac{1-\rho/4}{n}\mathbb{E}||X^t - \bar{X}^t||_F^2 + \frac{12\eta^2\zeta^2}{\rho} + 4\eta^2\sigma^2 + \frac{6\eta^2\mu^2}{\rho n}\mathbb{E}||B^t||_F^2.$$

*Proof.* Starting from the update step 15

$$\frac{1}{n}\mathbb{E}||X^{t+1} - \bar{X}^{t+1}||_F^2 = \frac{1}{n}\mathbb{E}||WX^t - \eta[G^t + \mu B^t] - (\bar{X}^t - \eta\bar{G}^t)||_F^2$$

$$= \frac{1}{n}\mathbb{E}||WX^t - \bar{X}^t - \eta(G^t - \bar{G}^t) - \eta\mu B^t||_F^2$$

$$\leq \frac{1}{n}\mathbb{E}||WX^t - \bar{X}^t - \eta(\mathbb{E}[G^t] - \mathbb{E}[\bar{G}^t]) - \eta\mu B^t||_F^2 + 4\eta^2\sigma^2$$

$$\overset{(a)}{\leq} \frac{1+\rho/2}{n}\mathbb{E}||WX^t - \bar{X}^t||_F^2 + \frac{\eta^2(1+2/\rho)}{n}\mathbb{E}||\mathbb{E}[G^t] - \mathbb{E}[\bar{G}^t] - \mu B^t||_F^2$$
$$+ 4\eta^2\sigma^2$$

$$\overset{(b)}{\leq} \frac{(1-\rho)(1+\rho/2)}{n}\mathbb{E}||X^t - \bar{X}^t||_F^2 + \frac{3\eta^2}{n\rho}\mathbb{E}||\mathbb{E}[G^t] - \mathbb{E}[\bar{G}^t] - \mu B^t||_F^2$$
$$+ 4\eta^2\sigma^2$$

$$\leq \frac{(1-\rho)(1+\rho/2)}{n}\mathbb{E}||X^t - \bar{X}^t||_F^2 + 4\eta^2\sigma^2 + \frac{6\eta^2}{n\rho}\mathbb{E}||\mathbb{E}[G^t] - \mathbb{E}[\bar{G}^t]||_F^2$$
$$+ \frac{6\eta^2\mu^2}{n\rho}\mathbb{E}||B^t||_F^2$$

$$\leq \frac{(1-\rho/2)}{n}\mathbb{E}||X^t - \bar{X}^t||_F^2 + 4\eta^2\sigma^2 + \frac{6\eta^2}{n\rho}\mathbb{E}||\mathbb{E}[G^t] - \nabla f(\bar{x}^t)||_F^2$$
$$+ \frac{6\eta^2\mu^2}{n\rho}\mathbb{E}||B^t||_F^2$$

$$\leq \frac{(1-\rho/2)}{n}\mathbb{E}||X^t - \bar{X}^t||_F^2 + 4\eta^2\sigma^2 + \frac{6\eta^2\mu^2}{n\rho}\mathbb{E}||B^t||_F^2$$
$$+ \frac{6\eta^2}{n\rho}\sum_{i=1}^{n}\mathbb{E}||\nabla f_i(x_i^t) \pm \nabla f_i(\bar{x}^t) - \nabla f(\bar{x}^t)||_F^2$$

$$\overset{(c)}{\leq} \frac{(1-\rho/2)}{n}\mathbb{E}||X^t - \bar{X}^t||_F^2 + 4\eta^2\sigma^2 + \frac{12\eta^2\zeta^2}{\rho} + \frac{6\eta^2\mu^2}{n\rho}\mathbb{E}||B^t||_F^2$$

$$+ \frac{12\eta^2}{n\rho}\sum_{i=1}^{n}\mathbb{E}||\nabla f_i(x_i^t) - \nabla f_i(\bar{x}^t)||_F^2$$

$$\overset{(d)}{\leq} \frac{(1-\rho/2)}{n}\mathbb{E}||X^t - \bar{X}^t||_F^2 + 4\eta^2\sigma^2 + \frac{12\eta^2\zeta^2}{\rho} + \frac{6\eta^2\mu^2}{n\rho}\mathbb{E}||B^t||_F^2$$

$$+ \frac{12\eta^2 L^2}{n\rho}\sum_{i=1}^{n}\mathbb{E}||x_i^t - \bar{x}^t||_F^2$$

$$= \Big(\frac{1-\rho/2}{n} + \frac{12\eta^2 L^2}{n\rho}\Big)\mathbb{E}||X^t - \bar{X}^t||_F^2 + 4\eta^2\sigma^2 + \frac{12\eta^2\zeta^2}{\rho}$$

$$+ \frac{6\eta^2\mu^2}{n\rho}\mathbb{E}||B^t||_F^2$$

$$\overset{(e)}{\leq} \frac{1-\rho/4}{n}\mathbb{E}||X^t - \bar{X}^t||_F^2 + 4\eta^2\sigma^2 + \frac{12\eta^2\zeta^2}{\rho} + \frac{6\eta^2\mu^2}{n\rho}\mathbb{E}||B^t||_F^2$$

(a) follows from the fact that $||a + b||^2 \leq (1+\alpha)||a||^2 + (1+\frac{1}{\alpha})||b||^2 \;\; \forall\alpha > 0$ and let $\alpha = \frac{\rho}{2}$. (b) uses $\mathbb{E}_W||ZW - \bar{Z}||_F^2 \leq (1-\rho)||ZW - \bar{Z}||_F^2$ and $1 + \frac{2}{\rho} \leq \frac{3}{\rho}$. (c) uses assumption-2 (7). (d) uses L-smoothness condition. (e) follows from the assumption that $\eta \leq \frac{\rho}{7L}$ $\hfill\square$

The next step is to find an upper bound for the bias term $\mathbb{E}||B^t||_F^2$.

**Lemma 4.** *Given assumptions 1-3 and $\frac{\mu}{1-\mu} \leq \frac{\rho}{42}$, we have*

$\frac{6\eta^2\mu^2}{\rho n(1-\mu)}\mathbb{E}||B^{t+1}||_F^2 \leq \Big(\frac{6\eta^2\mu^2}{\rho n(1-\mu)} - \frac{6\eta^2\mu^2}{\rho n}\Big)\mathbb{E}||B^t||_F^2 + \frac{\rho}{8n}\mathbb{E}||X^t - \bar{X}^t||_F^2 + \frac{\eta^2\zeta^2\rho}{8} + \frac{\eta^2\sigma^2\rho(1-\mu)}{8}.$

*Proof.* starting from the update step 15

$$B^{t+1} = -\frac{1}{\eta}[(2W - I)(X^{t+1} - X^t) + \eta G^t]$$

$$= -\frac{1}{\eta}[(2W - I)(WX^t - \eta G^t - \eta\mu B^t - X^t) + \eta G^t]$$

$$= -\frac{1}{\eta}[W(2W - I) - I]X^t + 2(W - I)G^t + \mu(2W - I)B^t.$$

Now,

$$\frac{1}{n}\mathbb{E}||B^{t+1}||_F^2 = \frac{1}{n}\mathbb{E}||-\frac{1}{\eta}(W(2W-I)-I)X^t + 2(W-I)G^t + \mu(2W-I)B^t||_F^2$$

$$= \frac{1}{n}\mathbb{E}||-\frac{1}{\eta}(W(2W-I)-I)X^t + 2(W-I)(G^t - \bar{G}^t) + \mu(2W-I)B^t||_F^2$$

$$= \frac{1}{n}\mathbb{E}||-\frac{1}{\eta}(W(2W-I)-I)X^t + 2(W-I)\mathbb{E}[G^t - \bar{G}^t] + \mu(2W-I)B^t||_F^2$$

$$+ \frac{1}{n}\mathbb{E}||2(W-I)(G^t - \mathbb{E}[G^t] - (\bar{G}^t - \mathbb{E}[\bar{G}^t]))||_F^2$$

$$\leq \frac{1}{n}\mathbb{E}||\frac{1}{\eta}(I - W(2W-I))X^t + 2(W-I)\mathbb{E}[G^t - \bar{G}^t] + \mu(2W-I)B^t||_F^2$$

$$+ 8\sigma^2$$

$$\overset{(a)}{\leq} \frac{1}{n}\Big(1 + \frac{1-\mu}{\mu}\Big)\mathbb{E}||\mu(2W-I)B^t||_F^2 + 8\sigma^2$$

$$+ \frac{1}{n}\Big(1 + \frac{\mu}{1-\mu}\Big)\mathbb{E}||\frac{1}{\eta}(I - W(2W-I))X^t + 2(W-I)\mathbb{E}[G^t - \bar{G}^t]||_F^2$$

$$\leq \frac{\mu}{n}\mathbb{E}||B^t||_F^2 + 8\sigma^2 + \frac{2}{n(1-\mu)}\mathbb{E}||\mathbb{E}[G^t - \bar{G}^t]||_F^2$$

$$+ \frac{2}{n\eta^2(1-\mu)}\mathbb{E}||(I - W(2W - I))X^t||_F^2$$

$$= \frac{1 - (1-\mu)}{n}\mathbb{E}||B^t||_F^2 + 8\sigma^2 + \frac{2}{n(1-\mu)}\mathbb{E}||\mathbb{E}[G^t - \bar{G}^t]||_F^2$$

$$+ \frac{2}{n\eta^2(1-\mu)}\mathbb{E}||(2W + I)(I - W)X^t||_F^2$$

$$\overset{(b)}{\leq} \frac{1 - (1-\mu)}{n}\mathbb{E}||B^t||_F^2 + 8\sigma^2 + \frac{2}{n(1-\mu)}\mathbb{E}||\mathbb{E}[G^t - \bar{G}^t]||_F^2$$

$$+ \frac{18}{n\eta^2(1-\mu)}\mathbb{E}||(I - W)X^t||_F^2$$

$$= \frac{1 - (1-\mu)}{n}\mathbb{E}||B^t||_F^2 + 8\sigma^2 + \frac{2}{n(1-\mu)}\mathbb{E}||\mathbb{E}[G^t - \bar{G}^t]||_F^2$$

$$+ \frac{18}{n\eta^2(1-\mu)}\mathbb{E}||(I - W)(X^t - \bar{X}^t)||_F^2$$

$$\leq \frac{1 - (1-\mu)}{n}\mathbb{E}||B^t||_F^2 + 8\sigma^2 + \frac{36}{n\eta^2(1-\mu)}\mathbb{E}||X^t - \bar{X})||_F^2$$

$$+ \frac{2}{n(1-\mu)}\mathbb{E}||\mathbb{E}[G^t] \pm \nabla f(\bar{x}^t) - \mathbb{E}[\bar{G}^t]||_F^2$$

$$\leq \frac{1 - (1-\mu)}{n}\mathbb{E}||B^t||_F^2 + 8\sigma^2 + \frac{36}{n\eta^2(1-\mu)}\mathbb{E}||X^t - \bar{X})||_F^2$$

$$+ \frac{8\zeta^2}{1-\mu} + \frac{4L^2}{n(1-\mu)}\mathbb{E}||X^t - \bar{X})||_F^2$$

$$= \frac{1 - (1-\mu)}{n}\mathbb{E}||B^t||_F^2 + 8\sigma^2 + \frac{4(9 + \eta^2 L^2)}{n\eta^2(1-\mu)}\mathbb{E}||X^t - \bar{X})||_F^2 + \frac{8\zeta^2}{1-\mu}$$

Multiplying both sides with $\frac{6\eta^2\mu^2}{\rho(1-\mu)}$

$$\frac{6\eta^2\mu^2}{n\rho(1-\mu)}\mathbb{E}||B^{t+1}||_F^2 \leq \Big(\frac{6\eta^2\mu^2}{n\rho(1-\mu)} - \frac{6\eta^2\mu^2}{n\rho}\Big)\mathbb{E}||B^t||_F^2 + \frac{24(9 + \eta^2 L^2)\mu^2}{n\rho(1-\mu)^2}\mathbb{E}||X^t - \bar{X}||_F^2$$

$$+ \frac{48\eta^2\mu^2\sigma^2}{\rho(1-\mu)} + \frac{48\eta^2\mu^2\zeta^2}{\rho(1-\mu)^2}$$

$$\overset{(c)}{\leq} \Big(\frac{6\eta^2\mu^2}{n\rho(1-\mu)} - \frac{6\eta^2\mu^2}{n\rho}\Big)\mathbb{E}||B^t||_F^2 + \frac{\rho}{8n}\mathbb{E}||X^t - \bar{X})||_F^2$$

$$+ \frac{\eta^2\rho\sigma^2(1-\mu)}{8} + \frac{\eta^2\rho\zeta^2}{8}$$

Note that $W - I < I$, $I - W < 2I$, $(W - I)\bar{X}^t = 0$ and $(W - I)\bar{G}^t = 0$. (a) follows from the fact that $||a + b||^2 \leq (1 + \alpha)||a||^2 + (1 + \frac{1}{\alpha})||b||^2 \; \forall \alpha > 0$ and let $\alpha = \frac{1-\mu}{\mu}$. (b) uses the fact that $||AB||_F^2 \leq \sigma_{max}^2(A)||B||_F^2$ where $A = 2W + I$, $B = (I - W)X^t$ and $\sigma_{max}^2(A) = 9$. (c) uses the assumption $\frac{\mu}{1-\mu} \leq \frac{\rho}{42}$ and $\eta \leq \frac{\rho}{7L}$. This implies that $\frac{24(9+\eta^2 L^2)\mu^2}{\rho(1-\mu)^2} \leq \frac{\rho}{8}$ and $\frac{48\mu^2}{\rho(1-\mu)^2} \leq \frac{\rho}{8}$ $\qquad\square$

We present the proof for Theorem 1 using Lemmas. 2, 3, and 4. Adding Lemmas. 3, and 4 and simplifying, we get

$$\frac{24L^2\eta}{n}\mathbb{E}||X^{t+1} - \bar{X}^{t+1}||_F^2 + \frac{144L^2\eta^3\mu^2}{n\rho^2(1-\mu)}\mathbb{E}||B^{t+1}||_F^2 \leq \left(\frac{24L^2\eta}{n} - \frac{3L^2\eta}{n}\right)\mathbb{E}||X^t - \bar{X}^t||_F^2$$
$$+ \frac{144L^2\eta^3\mu^2}{n\rho^2(1-\mu)}\mathbb{E}||B^t||_F^2 + \frac{312L^2\eta^3}{\rho^2}(\zeta^2 + \sigma^2(2-\mu)) \tag{17}$$

Finally, define a function $\Phi^t$ as shown below

$$\Phi^t = \frac{24L^2\eta}{n}\mathbb{E}||X^t - \bar{X}^t||_F^2 + \frac{144L^2\eta^3\mu^2}{n\rho^2(1-\mu)}\mathbb{E}||B^t||_F^2 + \mathbb{E}[f(\bar{x}^t) - f^*] \tag{18}$$

Now adding Lemma 2 and (17), we have the following

$$\Phi^{t+1} \leq \Phi^t - \frac{\eta}{4}\mathbb{E}||\frac{1}{n}\sum_{i=1}^n \nabla f_i(x_i^t)||^2 - \frac{\eta}{4}\mathbb{E}||\nabla f(\bar{x}^t)||^2 + \frac{L\eta^2\sigma^2}{n} + \frac{312L^2\eta^3}{\rho^2}(\zeta^2 + \sigma^2(2-\mu))$$

$$\leq \Phi^t - \frac{\eta}{4}\mathbb{E}||\nabla f(\bar{x}^t)||^2 + \frac{L\eta^2\sigma^2}{n} + \frac{312L^2\eta^3}{\rho^2}(\zeta^2 + \sigma^2(2-\mu))$$

$$\implies \frac{\eta}{4}\mathbb{E}||\nabla f(\bar{x}^t)||^2 \leq (\Phi^t - \Phi^{t+1}) + \frac{L\eta^2\sigma^2}{n} + \frac{312L^2\eta^3}{\rho^2}(\zeta^2 + \sigma^2(2-\mu))$$

Summing over t

$$\frac{1}{T}\sum_{t=0}^{T-1}\mathbb{E}||\nabla f(\bar{x}^t)||^2 \leq \frac{4}{\eta T}(f(\bar{x}^0 - f^*) + \eta\frac{4L\sigma^2}{n} + \eta^2\frac{1248L^2}{\rho^2}(\zeta^2 + \sigma^2(2-\mu)). \tag{19}$$

This concludes the proof of the Theorem 1.

### A.4 Proof of Corollary. 1

In the proof of Theorem 1, we assumed the following the following constraints on learning rate $\eta$ and scaling factor $\mu$:

$$(i) \quad \eta \leq \min\left\{\frac{1}{4L}, \frac{\rho}{7L}\right\}$$
$$(ii) \quad \frac{\mu}{1-\mu} \leq \frac{\rho}{42}.$$

We assume that the step size $\eta$ is $\mathcal{O}(\sqrt{\frac{n}{T}})$, where $n$ is the total number of agents and $T$ is the number of iterations. Given this assumption, we have the following order of each term in (19) of Theorem 1.

$$\frac{4}{\eta T}(f(\bar{x}^0 - f^*) = \mathcal{O}\left(\frac{1}{\sqrt{nT}}\right).$$

For the remaining terms we have,

$$\eta\frac{4L\sigma^2}{n} = \mathcal{O}\left(\frac{1}{\sqrt{nT}}\right), \quad \eta^2\frac{1248L^2}{\rho^2}(\zeta^2 + \sigma^2(2-\mu)) = \mathcal{O}\left(\frac{n}{T}\right).$$

Therefore, by omitting the constant $n$ in this context of higher order terms, there exists a constant $C > 0$ such that the overall convergence rate is as follows:

$$\frac{1}{T}\sum_{t=0}^{T-1}\mathbb{E}||\nabla f(\bar{x}^t)||^2 \leq C\left(\frac{1}{\sqrt{nT}} + \frac{1}{T}\right),$$

which suggests when $T$ is sufficiently large, *GUT* enables the convergence rate of $\mathcal{O}(\frac{1}{\sqrt{nT}})$.

# B   Algorithmic details

In this section, we present the pseudo-code for *QG-GUTm* which combines the proposed *GUT* algorithm with quasi-global momentum as Algorithm 2 and its PyTorch implementation version is shown in Algorithm 3. We also summarize the memory-efficient implementation of *GUT* in Algorithm 4.

---

**Algorithm 2** Global Update Tracking with momentum (*QG-GUTm*)

---

**Input:** Each agent $i \in [1, n]$ initializes model weights $x_i^{(0)}$ and neighbors' copy $\hat{x}_j^{(0)}$, momentum buffer $m_i^{(0)}$, step size $\eta$, momentum coefficient $\beta$, mixing matrix $W = [w_{ij}]_{i,j \in [1,n]}$, *GUT* scaling factor $\mu$, $I_{ij}$ are elements of $n \times n$ identity matrix, $\mathcal{N}(i)$ represents neighbors of $i$ including itself, and note $\hat{x}_i^t = x_i^t$.

Each agent simultaneously implements the TRAIN( ) procedure
1. **procedure** TRAIN( )
2.    **for** t=$0, 1, \ldots, T-1$ **do**
3.       $d_i^t \sim D_i$
4.       $g_i^t = \nabla_x f_i(d_i^t; \sum_{j \in \mathcal{N}(i)} w_{ij} * \hat{x}_j^t)$
5.       $\delta_i^t = g_i^t - \frac{1}{\eta} \sum_{j \in \mathcal{N}(i)} (w_{ij} - I_{ij}) * \hat{x}_j^t$
6.       $y_i^t = \delta_i^t + \mu \big[ \sum\limits_{j \in \mathcal{N}(i)} w_{ij}(m_j^{t-1} - \frac{1}{\eta}(\hat{x}_j^t - x_i^t)) - \delta_i^{t-1} \big]$
7.       $m_i^t = \beta m_i^{t-1} + (1 - \beta) y_i^t$
8.       SENDRECEIVE($m_i^t$)
9.       $x_i^{t+1} = x_i^t - \eta m_i^t$
10.      $\hat{x}_j^{t+1} = \hat{x}_j^t - \eta m_j^t \;\; \forall \;\; j \in N(i) \backslash i$
11.    **end**
12. **return**

---

---

**Algorithm 3** Global Update Tracking with momentum (*QG-GUTm*) – Pytorch Implementation

---

**Input:** Each agent $i \in [1, n]$ initializes model weights $x_i^{(0)}$ and neighbors' copy $\hat{x}_j^{(0)}$, momentum buffer $m_i^{(0)}$, step size $\eta$, momentum coefficient $\beta$, mixing matrix $W = [w_{ij}]_{i,j \in [1,n]}$, *GUT* scaling factor $\mu$, $I_{ij}$ are elements of $n \times n$ identity matrix, $\mathcal{N}(i)$ represents neighbors of $i$ including itself, and note $\hat{x}_i^t = x_i^t$.

Each agent simultaneously implements the TRAIN( ) procedure
1. **procedure** TRAIN( )
2.    **for** t=$0, 1, \ldots, T-1$ **do**
3.       $d_i^t \sim D_i$
4.       $g_i^t = \nabla_x f_i(d_i^t; \sum_{j \in \mathcal{N}(i)} w_{ij} * \hat{x}_j^t)$
5.       $\delta_i^t = g_i^t - \frac{1}{\eta} \sum_{j \in \mathcal{N}(i)} (w_{ij} - I_{ij}) * \hat{x}_j^t$
6.       $y_i^t = \delta_i^t + \mu \big[ \sum\limits_{j \in \mathcal{N}(i)} w_{ij}(m_j^{t-1} - \frac{1+\beta}{\eta}(\hat{x}_j^t - x_i^t)) - \delta_i^{t-1} \big]$
7.       $m_i^t = \beta m_i^{t-1} + y_i^t$
8.       SENDRECEIVE($m_i^t$)
9.       $x_i^{t+1} = x_i^t - \eta m_i^t$
10.      $\hat{x}_j^{t+1} = \hat{x}_j^t - \eta m_j^t \;\; \forall \;\; j \in N(i) \backslash i$
11.    **end**
12. **return**

---

**Algorithm 4** Global Update Tracking (Memory Efficient Implementation)

**Input:** Each agent $i \in [1, n]$ initializes model parameters $x_i^0$ and weighted model parameters of neighborhood $s_i^0$, step size $\eta$, *GUT* scaling factor $\mu$, mixing matrix $W = [w_{ij}]_{i,j \in [1,n]}$, $\mathcal{N}(i)$ represents neighbors of $i$ including itself.

Each agent simultaneously implements the TRAIN( ) procedure
1. **procedure** TRAIN( )
2.    **for** $t = 0, 1, \ldots, T - 1$ **do**
3.       $d_i^t \sim D_i$
4.       $g_i^t = \nabla_x F_i(s_i^t; d_i^t)$
5.       $\delta_i^t = g_i^t - \frac{1}{\eta}(s_i^t - x_i^t)$
6.       $y_i^t = \delta_i^t + \mu \left[ \sum_{j \in \mathcal{N}(i)} w_{ij} y_j^{t-1} - \frac{1}{\eta}(s_i^t - x_i^t) - \delta_i^{t-1} \right]$
7.       SENDRECEIVE($y_i^t$)
8.       $x_i^{t+1} = x_i^t - \eta y_i^t$
9.       $s_i^{t+1} = s_i^t - \eta \sum_{j \in \mathcal{N}(i)} w_{ij} y_j$
10.    **end**
11. **return** $\frac{1}{n} \sum_{i=1}^{n} x_i^T$

# C   Decentralized Learning Setup

For the decentralized setup, we use an undirected ring, undirected Dyck graph, and undirected torus graph topologies with a uniform mixing matrix. The undirected ring topology for any graph size has 3 peers per agent including itself and each edge has a weight of $\frac{1}{3}$. The undirected Dyck topology with 32 agents has 4 peers per agent including itself and each edge has a weight of $\frac{1}{4}$. The undirected torus topology with 32 agents has 5 peers per agent including itself and each edge has a weight of $\frac{1}{5}$. All our experiments were conducted on a system with Nvidia GTX 1080ti card with 4 GPUs except for ImageNette simulations. We used NVIDIA A40 card with 4 GPUs for ImageNette simulations.

## C.1   Datasets

In this section, we give a brief description of the datasets used in our experiments. We use a diverse set of datasets each originating from a different distribution of images to show the generalizability of the proposed techniques.

**CIFAR-10:** CIFAR-10 [13] is an image classification dataset with 10 classes. The image samples are colored (3 input channels) and have a resolution of $32 \times 32$. There are $50,000$ training samples with $5000$ samples per class and $10,000$ test samples with $1000$ samples per class.

**CIFAR-100:** CIFAR-100 [13] is an image classification dataset with 100 classes. The image samples are colored (3 input channels) and have a resolution of $32 \times 32$. There are $50,000$ training samples with $500$ samples per class and $10,000$ test samples with $100$ samples per class. CIFAR-100 classification is a harder task compared to CIFAR-10 as it has 100 classes with very few samples per class to learn from.

**Fashion MNIST:** Fashion MNIST [25] is an image classification dataset with 10 classes. The image samples are in greyscale (1 input channel) and have a resolution of $28 \times 28$. There are $60,000$ training samples with $6000$ samples per class and $10,000$ test samples with $1000$ samples per class.

**Imagenette:** Imagenette [10] is a 10-class subset of the ImageNet dataset. The image samples are colored (3 input channels) and have a resolution of $224 \times 224$. There are $9469$ training samples with roughly $950$ samples per class and $3925$ test samples.

## C.2   Network Architecture

We replace ReLU+BatchNorm layers of all the model architectures with EvoNorm-S0 as it was shown to be better suited for decentralized learning over non-IID distributions.

**VGG-11:** We modify the standard VGG-11 [21] architecture by reducing the number of filters in each convolutional layer by $4\times$ and using only one dense layer with 128 units. Each convolutional layer is followed by EvoNorm-S0 as the activation-normalization layer. VGG-11 has $0.58M$ trainable parameters.

**ResNet-20:** For ResNet-20 [8], we use the standard architecture with $0.27M$ trainable parameters except that BatchNorm+ReLU layers are replaced by EvoNorm-S0.

**LeNet-5:** For LeNet-5 [14], we use the standard architecture with $61,706$ trainable parameters.

**MobileNet-V2:** We use the the standard MobileNet-V2 [20] architecture used for CIFAR dataset with $2.3M$ parameters except that BatchNorm+ReLU layers are replaced by EvoNorm-S0.

## C.3  Hyper-parameters

This section presents a detailed description of the hyper-parameters used in our experiments. All the experiments were run for three randomly chosen seeds. We decay the step size by 10x after 50% and 75% of the training, unless mentioned otherwise. The hyper-parameter $\mu$ is set to 0.9 for all the experiments using *GUT* optimizer. We used grid search to choose the hyper-parameter $\mu$ for *QG-GUTm*.

**Hyper-parameters for experiments in Table 1:** All the experiments have the stopping criteria set to 200 epochs. The initial learning rate is set to 0.1. We decay the step size by $10\times$ in multiple steps at $100^{th}$ and $150^{th}$ epoch. Table 6 presents values of the scaling factor $\mu$ used in the experiments. For all the experiments, we use a mini-batch size of 32 per agent. The stopping criteria is a fixed number of epochs. We have used a momentum of 0.9 for all QG-DSDm and *QG-GUTm* experiments.

Table 6: The value of scaling factor $\mu$ used for training CIFAR-10 with non-IID data using ResNet-20 and VGG-11 model architectures presented in Table 1

| Agents ($n$) | Method | ResNet-20 | | |
|---|---|---|---|---|
| | | $\alpha = 1$ | $\alpha = 0.1$ | $\alpha = 0.01$ |
| | DSGD | 0.0 | 0.0 | 0.0 |
| | *GUT (ours)* | 0.9 | 0.9 | 0.9 |
| 16 | QG-DSGDm | 0.0 | 0.0 | 0.0 |
| | *QG-GUTm (ours)* | 0.04 | 0.06 | 0.04 |
| | DSGD | 0.0 | 0.0 | 0.0 |
| | *GUT (ours)* | 0.9 | 0.9 | 0.9 |
| 32 | QG-DSGDm | 0.0 | 0.0 | 0.0 |
| | *QG-GUTm (ours)* | 0.04 | 0.04 | 0.04 |
| Agents ($n$) | Method | VGG-11 | | |
| | | $\alpha = 1$ | $\alpha = 0.1$ | $\alpha = 0.01$ |
| | DSGD | 0.0 | 0.0 | 0.0 |
| | *GUT (ours)* | 0.9 | 0.9 | 0.9 |
| 16 | QG-DSGDm | 0.0 | 0.0 | 0.0 |
| | *QG-GUTm (ours)* | 0.06 | 0.08 | 0.09 |
| | DSGD | 0.0 | 0.0 | 0.0 |
| | *GUT (ours)* | 0.9 | 0.9 | 0.9 |
| 32 | QG-DSGDm | 0.0 | 0.0 | 0.0 |
| | *QG-GUTm (ours)* | 0.08 | 0.08 | 0.08 |

**Hyper-parameters for experiments in Table 2:** All the experiments have the stopping criteria set to 200 epochs. The initial learning rate is set to 0.1. We decay the step size by $10\times$ in multiple steps at $100^{th}$ and $150^{th}$ epoch. Table 7 presents values of the scaling factor $\mu$ used in the experiments. For all the experiments, we use a mini-batch size of 32 per agent. The stopping criteria is a fixed number of epochs. For all QG-DSDm and *QG-GUTm* experiments, we have used a momentum of 0.9 and Nesterov is set to False. We did not use any regularization in our experiments on non-IID data i.e., weight decay is set to zero. We set the weight decay to be $1e^{-4}$ for experiments on IID data (DSGDm case in Table. 1).

**Hyper-parameters for experiments in Table 3:** All the experiments with Fashion-MNIST and Imagenette datasets have the stopping criteria set to 100 epochs where as CIFAR-100 experiments

Table 7: The value of scaling factor $\mu$ used for training CIFAR-10 with non-IID data using ResNet-20 model architecture over varies graph topologies presented in Table 2

| Method | Dyck Graph (32 agents) | | Torus (32 agents) | |
|---|---|---|---|---|
| | $\alpha = 0.1$ | $\alpha = 0.01$ | $\alpha = 0.1$ | $\alpha = 0.01$ |
| QG-DSGDm | 0.0 | 0.0 | 0.0 | 0.0 |
| QG-GUTm | 0.05 | 0.05 | 0.05 | 0.05 |

have the stopping criteria as 200 epochs. The initial learning rate is set to 0.1 for experiments on CIFAR-100 and Fashion MNIST datasets. The initial learning rate is set to 0.01 for experiments on the Imagenette dataset. We decay the step size by $10\times$ in multiple steps at $50^{th}$ and $75^{th}$ epoch. Table 8 presents values of the scaling factor $\mu$ used in the experiments. For all the experiments, we use a mini-batch size of 32 per agent. The stopping criteria is a fixed number of epochs. We have used a momentum of 0.9 for all QG-DSDm and *QG-GUTm* experiments.

Table 8: The value of scaling factor $\mu$ used for training different datasets over 16 agents ring topology presented in Table 3

| Method | Fashion MNIST (LeNet-5) | | CIFAR-100 (ResNet-20) | | Imagenette (MobileNet-V2) | |
|---|---|---|---|---|---|---|
| | $\alpha = 0.1$ | $\alpha = 0.01$ | $\alpha = 0.1$ | $\alpha = 0.01$ | $\alpha = 0.1$ | $\alpha = 0.01$ |
| QG-DSGDm | 0.0 | 0.0 | 0.0 | 0.0 | 0.0 | 0.0 |
| *QG-GUTm* | 0.01 | 0.005 | 0.005 | 0.005 | 0.03 | 0.04 |

# D   Additional Results

Table. 9 shows that the quasi-global variant of Global Update Tracking surpasses the other existing methods even for a higher degree of heterogeneity i.e., $\alpha = 0.01$. We also observe that the Nesterov momentum hurts the performance when the heterogeneity in the data distribution is high. Table. 10 compares the DSGDm baseline with the proposed *GUT* and *QG-GUTm*. It shows that for a higher degree of heterogeneity, *GUT* outperforms even the momentum version of DSGD. Table. 11 evaluates the proposed method on various datasets trained on same model architecture (ResNet-20).

Table 9: Evaluating Global Update Tracking (GUT) with various versions of momentum using CIFAR-10 dataset trained on ResNet-20 architecture over 16 agents ring topology for $\alpha = 0.01$.

| Method | Local Momentum | Nesterov | Quasi-Global Momentum | Global Update Tracking | Test Accuracy $\alpha = 0.01$ |
|---|---|---|---|---|---|
| DSGD | x | x | x | x | $54.66 \pm 4.74$ |
| DSGDm | ✓ | x | x | x | $65.62 \pm 4.95$ |
| DSGDm-N | ✓ | ✓ | x | x | $63.66 \pm 4.44$ |
| QG-DSGDm | x | x | ✓ | x | $79.85 \pm 2.13$ |
| QG-DSGDm-N | x | ✓ | ✓ | x | $78.64 \pm 2.14$ |
| GUT | x | x | x | ✓ | $70.16 \pm 4.94$ |
| GUTm | ✓ | x | x | ✓ | $64.25 \pm 5.31$ |
| GUTm-N | ✓ | ✓ | x | ✓ | $63.42 \pm 2.74$ |
| QG-GUTm | x | x | ✓ | ✓ | $\mathbf{81.04} \pm 1.66$ |
| QG-GUTm-N | x | ✓ | ✓ | ✓ | $80.09 \pm 3.82$ |

# E   Hardware Efficiency

In this section, we present the quantitative results on communication, memory, and compute overheads of the various decentralized algorithms. The communication cost incurred by the proposed *GUT*

Table 10: Average test accuracy of different decentralized algorithms evaluated on CIFAR-10, distributed with different degrees of heterogeneity (non-IID) for various models over ring topologies.

| Agents ($n$) | Method | ResNet-20 | | |
| | | $\alpha = 1$ | $\alpha = 0.1$ | $\alpha = 0.01$ |
|---|---|---|---|---|
| 16 | *GUT (ours)* | $84.72 \pm 0.20$ | $81.86 \pm 1.99$ | $70.16 \pm 4.94$ |
| | DSGDm [15] | $86.60 \pm 0.54$ | $79.87 \pm 1.73$ | $65.62 \pm 4.95$ |
| | *QG-GUTm (ours)* | $\mathbf{88.22} \pm 0.36$ | $\mathbf{86.44} \pm 0.36$ | $\mathbf{81.04} \pm 1.66$ |
| 32 | *GUT (ours)* | $79.24 \pm 0.33$ | $76.07 \pm 0.23$ | $60.72 \pm 1.03$ |
| | DSGDm [15] | $86.12 \pm 0.32$ | $77.43 \pm 1.71$ | $52.82 \pm 4.02$ |
| | *QG-GUTm (ours)* | $\mathbf{87.48} \pm 0.33$ | $\mathbf{84.94} \pm 0.60$ | $\mathbf{72.04} \pm 3.18$ |

| Agents ($n$) | Method | VGG-11 | | |
| | | $\alpha = 1$ | $\alpha = 0.1$ | $\alpha = 0.01$ |
|---|---|---|---|---|
| 16 | *GUT (ours)* | $82.12 \pm 0.09$ | $81.24 \pm 0.95$ | $76.62 \pm 1.37$ |
| | DSGDm [15] | $81.77 \pm 0.38$ | $74.20 \pm 1.89$ | $58.44 \pm 14.58$ |
| | *QG-GUTm (ours)* | $\mathbf{84.46} \pm 0.33$ | $\mathbf{83.05} \pm 0.48$ | $\mathbf{78.32} \pm 1.03$ |
| 32 | *GUT (ours)* | $80.37 \pm 0.33$ | $79.55 \pm 1.00$ | $73.59 \pm 1.26$ |
| | DSGDm [15] | $81.89 \pm 0.29$ | $74.73 \pm 0.73$ | $61.60 \pm 2.80$ |
| | *QG-GUTm (ours)* | $\mathbf{84.32} \pm 0.11$ | $\mathbf{83.39} \pm 0.38$ | $\mathbf{77.41} \pm 3.44$ |

Table 11: Average test accuracy of different decentralized algorithms evaluated with various datasets trained on ResNet-20 architecture, distributed with different degrees of heterogeneity over 16 agents ring topology.

| Method | Fashion MNIST | | CIFAR-100 | | Imagenette | |
| | $\alpha = 0.1$ | $\alpha = 0.01$ | $\alpha = 0.1$ | $\alpha = 0.01$ | $\alpha = 0.1$ | $\alpha = 0.01$ |
|---|---|---|---|---|---|---|
| DSGDm [15] | $87.89 \pm 2.34$ | $79.41 \pm 3.29$ | $47.93 \pm 1.69$ | $42.57 \pm 2.71$ | $66.89 \pm 3.12$ | $47.87 \pm 4.03$ |
| QG-DSGDm [16] | $92.21 \pm 0.01$ | $90.59 \pm 0.92$ | $53.19 \pm 1.68$ | $44.17 \pm 3.64$ | $73.93 \pm 2.01$ | $56.30 \pm 5.43$ |
| *QG-GUTm* | $\mathbf{92.55} \pm 0.16$ | $\mathbf{91.70} \pm 0.36$ | $\mathbf{53.40} \pm 1.23$ | $\mathbf{50.45} \pm 1.30$ | $\mathbf{75.44} \pm 2.22$ | $\mathbf{57.47} \pm 5.33$ |

and *QG-GUTm* methodologies is the same as the DSGD or QG-DSGDm techniques. All the above-mentioned algorithms communicate a vector with the size same as the model parameters. However, gradient tracking incurs $2\times$ communication cost in terms of model size. The communication cost and memory requirements for all the experiments are reported in Table. 12 and 13.

We report the numbers for communication, memory, and compute overheads for the memory-efficient implementation of GUT Algorithm. 4 in Table. 14. Memory overhead is reported as the percentage of additional memory required per agent during training with a batch size of 32.

$$\text{Memory overhead} = \frac{\text{Additional memory due to } GUT}{\text{Total Memory}}$$

The total memory includes the memory required to store model parameters, activations, gradients, gossip buffer, tracking variable ($y_i$), and weighted neighbors' parameters ($s_i$). We observe that for compact models such as ResNet and MobileNet, the memory overhead is less than $2\%$. However, for larger models such as VGG-11, the memory overhead shoots up to $14\%$. The computational overhead is reported as the percentage of additional FLOPs required per sample per agent during training.

$$\text{Compute overhead} = \frac{\text{Additional compute due to } GUT}{\text{Total Compute}}$$

The total compute includes the forward pass, backward pass, model updates, gossip averaging, and tracking variable computation flops. We observe that for compact models such as ResNet and MobileNet, the compute overhead is around $2\%$. However, for larger models such as VGG-11, the compute overhead shoots up to $15\%$.

Table 12: Communication cost and memory requirement per agent during training of CIFAR-10 dataset on various model architectures and graph topologies.

| Model | Graph | Graph Size | Method | Comm. Cost (GB) | Memory (MB) |
|---|---|---|---|---|---|
| ResNet-20 | Ring | 16 | Gradient Tracking | 83.66 | 129.60 |
| | | | QG-DSGDm | 41.83 | 128.56 |
| | | | *QG-GUTm* | 41.83 | 130.64 |
| ResNet-20 | Ring | 32 | Gradient Tracking | 41.93 | 129.60 |
| | | | QG-DSGDm | 20.97 | 128.56 |
| | | | *QG-GUTm* | 20.97 | 130.64 |
| ResNet-20 | Torus | 32 | Gradient Tracking | 83.86 | 129.60 |
| | | | QG-DSGDm | 41.93 | 128.56 |
| | | | *QG-GUTm* | 41.93 | 130.64 |
| ResNet-20 | Dyck | 32 | Gradient Tracking | 62.90 | 129.60 |
| | | | QG-DSGDm | 31.45 | 128.56 |
| | | | *QG-GUTm* | 31.45 | 130.64 |
| VGG-11 | Ring | 16 | Gradient Tracking | 178.02 | 31.84 |
| | | | QG-DSGDm | 89.01 | 29.63 |
| | | | *QG-GUTm* | 89.01 | 34.05 |
| VGG-11 | Ring | 32 | Gradient Tracking | 89.22 | 31.84 |
| | | | QG-DSGDm | 44.61 | 29.63 |
| | | | *QG-GUTm* | 44.61 | 34.05 |

Table 13: Communication cost and memory requirement incurred per agent during training of various dataset and model architectures over 16 agents ring topology.

| Dataset | Model | Method | Comm. Cost (GB) | Memory (MB) |
|---|---|---|---|---|
| Fashion MNIST | LeNet-5 | Gradient Tracking | 11.42 | 4.86 |
| | | QG-DSGDm | 5.71 | 4.62 |
| | | *QG-GUTm* | 5.71 | 5.10 |
| CIFAR-100 | ResNet-20 | Gradient Tracking | 85.46 | 129.72 |
| | | QG-DSGDm | 42.73 | 128.66 |
| | | *QG-GUTm* | 42.73 | 130.78 |
| Imagenette | MobileNet-V2 | Gradient Tracking | 68.86 | 3774 |
| | | QG-DSGDm | 34.43 | 3765 |
| | | *QG-GUTm* | 34.43 | 3782 |

Table 14: Communication, memory, and compute overhead incurred per agent during training of various datasets and model architectures for the proposed *GUT* algorithm. Note that the overheads are independent of the graph topology and graph size.

| Dataset | Model | Communication Overhead | Memory Overhead | Compute Overhead |
|---|---|---|---|---|
| Fashion MNIST | LeNet-5 | 0.00 | 0.099 | 0.275 |
| CIFAR-10 | ResNet-20 | 0.00 | 0.016 | 0.021 |
| CIFAR-10 | VGG-11 | 0.00 | 0.138 | 0.149 |
| CIFAR-100 | ResNet-20 | 0.00 | 0.016 | 0.022 |
| Imagenette | MobileNet-V2 | 0.00 | 0.005 | 0.021 |

