# OpenReview forum: "Global Update Tracking: A Decentralized Learning Algorithm for Heterogeneous Data"
_NeurIPS.cc/2023/Conference — NeurIPS 2023 poster_

### Official Review · Reviewer_b7eV · 2023-07-04

**Soundness:** 3 good
**Presentation:** 3 good
**Contribution:** 2 fair
**Rating:** 5
**Confidence:** 3

**Summary:**

This work proposes a novel decentralized learning algorithm based on gradient tracking mechanism, called Global Update Tracking (GUT), which aims to mitigate the impact of heterogeneous data distribution. The proposed GUT algorithm overcomes the bottleneck of communication overhead by allowing agents to store a copy of their neighbors’ model parameters and then tracking the model updates instead of the gradients. Numerous experiments have proven that the proposed Global Update Tracking with Quasi-Global momentum (QG-GUTm) outperforms the current state-of-the-art decentralized learning algorithm on a spectrum of heterogeneous data.

**Strengths:**

1.	This paper proposes a powerful global update tracking method and achieves good performance on multiple benchmarks.
2.	The authors provide a detailed sensitivity study of the method's performance with respect to model architecture and hyper-parameters.
3.	The paper is well written and easy for the reader to read and understand.


**Weaknesses:**

1.	In the experimental part, the quantitative experiments about communication parameters(cost) are not clear enough. Can the authors provide specific quantitative results for comparison so that readers can easily understand them?

**Questions:**

Please refer to the weaknesses section.

**Limitations:**

This paper does not have any specific societal impact, other than the ones associated with other algorithms.

---

> ### Author Rebuttal · Authors · 2023-08-07
>
> We thank the reviewer for their time and feedback.
> The following Table 1 provides the communication cost for the experiments on 16 agents ring topology measured in terms of the total amount of data transferred in GB during training per agent. The communication costs for the remaining experiments are presented in **Table R3** of the rebuttal pdf.
>
> Table 1: Communication cost per agent for training various datasets on different model architectures over 16 agents ring topology.
>  | Dataset   | Model |  Method| Communication Cost (GB) |
> | ------------------ | ----------------- | ------------ |  :--------: |
> |CIFAR-10 | ResNet-20   |Gradient Tracking| 83.66|
> |CIFAR-10 | ResNet-20  |QG-DSGDm| 41.83|
> |CIFAR-10 | ResNet-20  |QG-GUTm| 41.83|
> |CIFAR-100 | ResNet-20  |Gradient Tracking| 85.46|
> |CIFAR-100 | ResNet-20  |QG-DSGDm| 42.73|
> |CIFAR-100 | ResNet-20  |QG-GUTm| 42.73|
> |Fashion MNIST | LeNet-5  |Gradient Tracking| 11.42|
> |Fashion MNIST | LeNet-5  |QG-DSGDm| 5.71|
> |Fashion MNIST | LeNet-5  |QG-GUTm| 5.71|
> |ImageNette | MobileNet-V2 |Gradient Tracking| 68.86|
> |ImageNette | MobileNet-V2  |QG-DSGDm| 34.43|
> |ImageNette | MobileNet-V2  |QG-GUTm| 34.43|
>
> The communication cost for the proposed GUT or QG-GUTm algorithm is the same as DSGD or QG-DSGDm methods i.e., the proposed methods have no additional communication overhead.
>
> We have answered all the questions raised by the reviewer and would be happy to answer any further questions.

---

### Official Review · Reviewer_tQhr · 2023-07-05

**Soundness:** 2 fair
**Presentation:** 2 fair
**Contribution:** 2 fair
**Rating:** 5
**Confidence:** 1

**Summary:**

The performance of decentralized learning is limited to the different distribution over devices. To address this issue, this paper proposes a method that is less susceptible to variations in data distribution, named GUT. The proposed GUT tracks the global/average model updates. Experiments show GUT achieves good performance on some benchmarks with various models. The paper also analyzes the convergence of the proposed method.

**Strengths:**

1) The issue of heterogeneous data on devices in decentralized learning is critical, the paper shows a good motivation to address this.
2) On the given benchmarks and models, the proposed method seems to perform well.
3) The analysis of convergence of the method is detailed.

**Weaknesses:**

1) There are too many descriptions of related work in the introduction, making it difficult for readers to find the gist.

2) The analysis of method strengths is not clear enough. Why the proposed method is better than previous methods? Why tracking the model updates instead of the gradients can save twice the communication overhead? Can differences in the data distribution of all agents be mitigated by communicating with neighboring agents?

3) References to employed benchmarks (CIFAR-10, CIFAR-100, Fashion MNIST, and Imagenette) and models (VGG-11, ResNet-20, LeNet-5 and, MobileNet-V2) are missed.

4) The evaluated benchmarks are relatively small. More large-scale benchmarks should be supplemented, such as ImageNet.

5) Some larger and state-of-the-art models should be evaluated to verify the effectiveness of the proposed method, such as ResNet-50, ResNet-100, ViT.

6) The evaluations on Fashion MNIST, CIFAR-100, Imagenette employ different model. The models should be unified.

7) Since one of the main contributions of the method is to reduce overhead, the article lacks a quantitative analysis of overhead.

**Questions:**

In rebuttal, the reviewer hope to see the response to 2),4)-7) in weaknesses.

**Limitations:**

The paper provides a compelling discussion about the limitation.

---

> ### Author Rebuttal · Authors · 2023-08-07
>
> We thank the reviewer for their time and feedback. We answer each of the questions raised in the weakness section here.
> 1. We will update the literature review to only include decentralized learning works on heterogeneous data.
> 2.  The difference in data distribution across agents results in a huge variation in the local gradients and hence results in poor performance of decentralized learning. These differences can not be completely mitigated by communicating with neighboring agents. However, we can reduce this variance in local gradients through mechanisms such as gradient tracking. In these techniques, the key idea is to track the global/averaged gradient through a tracking variable. The tracking variable will have less variance across agents than the local gradient and is closer to the global gradient. But tracking mechanisms have a communication overhead of 2x as they communicate the tracking variable along with the model parameters to/from the neighbors at every iteration. Note that model parameters from neighbors are required for the gossip averaging step and tracking variables of the neighbors for updating the current agent’s tracking variable (bias correction). The goal of this work is to design an algorithm similar to gradient tracking but without communication overhead. To achieve this, every agent i communicates model updates i.e., $x_j^{t}-x_j^{t-1}$ which is the same as the tracking variable. Now, agent $i$ can recover the neighbor’s model parameters $x_j^t$ from the model updates by adding them to the local copy of $x_j^{t-1}$. The recovered neighbors’ parameters can then be used for the gossip averaging step. As for the tracking part, the model updates $x_i^{t}-x_i^{t-1}$ include two components -- (a) the gradient update $g_i^t$ and (b) the gossip update  $\sum\limits_{j \in N(i)}w_{ij} * (x_j^t -x_i^t)$.
> Ideally, we would like $g_i^t$ to be the average gradient i.e., $\frac{1}{n}\sum\limits_jg_j^t$ and $\sum\limits_{j \in N(i)}w_{ij} * (x_j^t -x_i^t)$ to be $\frac{1}{n}\sum\limits_n(x_j^t -x_i^t)$ -- as if the graph is fully connected. This implies that tracking the model updates not only inherently tracks the global gradient but also tracks the ideal gossip update. Hence the proposed GUT performs better than the previous methods at no additional communication overhead. Note that $n$ is the total number of agents and $\mathcal{N}(i)$ is the neighbors of agent $i$. We hope we clarified the contributions of the proposed methods and would be happy to provide more details if needed.
> 3. The reference for all the benchmark datasets and models are presented in Appendix C. We will add them to the main paper as well.
> 4. We provide the results on the ImageNet dataset trained on ResNet-18 architecture over 16 agents ring topology with alpha = 1 and 0.1 in below Table 1. We use an initial learning rate of 0.1 and decay it by a factor of 10 at epochs 15 and 37. The stopping criterion is set to 50 epochs.
>
> Table 1: Average test accuracy of decentralized algorithms evaluated on ImageNet with ResNet-18 over 16 agents ring.
> |Method|Test ACC ($\alpha=1$)| Test Acc ($\alpha=0.1$)|
> |-------------|-------------|-------------|
> |DSGD|$53.15$|$45.09$|
> |*GUT* (ours)|$53.57$|$46.33$|
> |QG-DSGDm|$60.85$|$57.17$|
> |*QG-GUTm* (ours)|$60.88$|$57.85$|
>
> 5.  Our current computational power cannot handle large ViT models. However, we do present experiments on Mobilenet-V2 which is more suitable for the edge applications.
> 6. Table 3 of the main paper uses the standard models that are usually employed for the respective datasets. We provide the results on ResNet-20 for all the datasets in the below table 2
>
> Table 2: Test accuracy of decentralized algorithms evaluated on ResNet-20 over 16 agents ring.
> |Method|Fashion MNIST|Fashion MNIST| CIFAR-100| CIFAR-100| Imagenette|Imagenette|
> |-------------|-------------|-------------|-------------|-------------|-------------|-------------|
> | | $\alpha=0.1$ | $\alpha=0.01$ | $\alpha=0.1$ | $\alpha=0.01$ | $\alpha=0.1$ | $\alpha=0.01$|
>  |DSGDm  | $87.89 \pm 2.34$ | $79.41 \pm 3.29$| $47.93 \pm 1.69$ | $42.57 \pm 2.71$| $66.89 \pm 3.12$ | $47.87 \pm 4.03$|
>  |QG-DSGDm  | $92.21 \pm 0.01$ | $90.59 \pm 0.92$ | $ 53.19 \pm 1.68 $ | $ 44.17 \pm 3.64 $ | $73.93 \pm  2.01$ | $56.30 \pm  5.43$|
> | *QG-GUTm* (ours) | $\mathbf{92.55} \pm 0.16$| $\mathbf{91.70} \pm 0.36$ | $ \mathbf{53.40} \pm 1.23 $ | $ \mathbf{50.45} \pm 1.30 $ | $\mathbf{75.44} \pm 2.22$ | $ \mathbf{57.47} \pm 5.33$|
>
> 7. We provide the quantitative results on communication cost and memory requirements of training for all the experiments in **Table R3** in the attached rebuttal pdf. We also provide the compute and memory overheads in **Table R4** in the attached rebuttal pdf. In Table R4, memory overhead is reported as the fraction of additional memory required per agent during training with a batch size of 32 per agent, and the computational overhead is reported as the fraction of additional FLOPs required per sample per agent during training.
>
> $\text{Memory overhead} = \frac{\text{Additional memory due to GUT}} {\text{Total memory}}$
>
> $\text{Compute overhead} = \frac{\text{Additional compute due to GUT}} {\text{Total compute}}$
>
> We have answered all the questions raised by the reviewer and would be happy to answer any further questions.

---

> > ### Comment · Reviewer_tQhr · 2023-08-16
> > **Re: Official Review of Submission597 by Reviewer tQhr and Rebuttal**
> >
> > 1. Thanks for the authors' rebuttal, I have updated my rating due to their rebuttal and other reviewers' comments;
> >
> > 2. I must say that this submission is out of my scope and encourage the AC consider this issue when they make decision.

---

### Official Review · Reviewer_NsAw · 2023-07-06

**Soundness:** 3 good
**Presentation:** 2 fair
**Contribution:** 3 good
**Rating:** 6
**Confidence:** 3

**Summary:**

The paper proposes a new decentralized learning algorithm called Global Update Tracking (GUT) that can mitigate the impact of heterogeneous data in decentralized learning without introducing any communication overhead. The proposal was demonstrated 4 computer vision datasets.

**Strengths:**

The paper proposes an important and useful application for training deep learning models in decentralized way. There are many strengths of the proposal including the following:

1. GUT is less susceptible to variations in data distribution across devices as shown by different experiments and ablation study

2. The performance is better than DSGD and DSGDm. However, it is shown only for CIFAR-10.

**Weaknesses:**

There are several questions:

1. The paper claims that GUT does not introduce any communication overhead but no results are shown to justify that. However using Figure 2, authors have claimed that their method converges faster than quasi global gossip. The comparison/improvement is not clear from the figure.

2. The author claim that agents in GUT store copy of their neighbors. How often this copy is updated? What is the communication overhead in that?

3. Performance comparison with related methods is not shown on all 4 datasets.

4. The data partition is fixed, non-overlapping and never shuffled across agents during training. How does this partition is defined?

**Questions:**

Please see weaknesses.

**Limitations:**

There are several limitations that the authors should address:

1. The data partition across agents is not clearly explained. Necessary details for all datasets is required. For example, number of samples per class per agent. Implementation details are also missing from the main paper.

2. Detailed comparison with the related methods is missing. For example, Table 3 compare performance on 3 datasets but no related method is shown.

---

> ### Author Rebuttal · Authors · 2023-08-07
>
> We thank the reviewer for their time and feedback. We answer each of the questions raised in the weakness section here.
> 1. (a) GUT/QG-GUTm communicates model updates $(x^{t}-x^{t-1})$ whereas DSGD/QG-DSGDm communicates model parameters $x^{t}$. Both these vectors are of the same size i.e., model size. We present the communication cost in terms of the data transferred per agent during training in **Table.R3** in the attached rebuttal pdf.
> (b) Figure 2 plots the consensus error for a simple gossip averaging task with varying graph sizes (spectral gap). We compare gossip with GUT (dotted blue curve) with simple gossip (dotted red curve). The figure shows that gossip with GUT reaches a lower consensus error than simple gossip. We also plot the consensus error for the quasi-momentum version of both algorithms. Simple gossip with QGM is shown by the solid red curve and GUT with QGM is shown by the solid blue curve. Figure.2(c) illustrates that for graphs with a smaller spectral gap (which corresponds to more agents), the proposed QG-GUTm can converge faster than quasi-global gossip (gossip with QGM).
> 2. The neighbors’ copy is updated in every iteration (line 9 in Algorithm 1) and does not require additional communication. The main difference between DGSD and GUT is the type of information that the agents communicate but both algorithms communicate vectors of the same size. In the DSGD algorithm, a given agent $i$ receives the model parameters $x_j^t$ at each iteration from its neighbors and uses it for gossip averaging. Whereas in GUT,  a given agent $i$ receives the model updates $(x_j^{t}-x_j^{t-1})$ at each iteration from its neighbors and recovers $x_j^{t}$ by adding the received model update $(x_j^{t}-x_j^{t-1})$ to the stored copy of $x_j^{t-1}$. Now we use the recovered $x_j^t$ for gossip averaging and model updates for bias computation or tracking.
> 3. We compare CIFAR-10 with DSGD, DGSDm, and QG-DSGDm. However, for the remaining datasets in Table. 3, we compare with QG-DSGDm as it is the existing state-of-the-art baseline for heterogenous data with no communication overhead. For more baselines, we report DSGDm along with QG-DGSDm in **Table.R2** in the attached rebuttal pdf.
> 4. At the beginning of the training, we divide the dataset across the agents where the label distribution follows the Dirichlet distribution and the non-iidness or the skew is controlled by the factor $\alpha$. The dataset is sampled without replacement and hence is non-overlapping across the agents i.e., each agent has a different part of the training dataset. We don't shuffle the dataset across the agents during the training. The visualization of the CIFAR-10 dataset with varying degrees of alpha on a 16-agent ring topology can be found in Figure.1,8 of [1]. We will add similar bubble plots for all our experiments in the appendix.
>
> We answer each of the questions raised in the Limitations section here.
> 1. For the interest of the space in the main paper, we added the implementation details of our experiments in Appendix C. We will include the class distribution across agents in the appendix and also add bubble plots for visualization.
> 2. We compared the results in Table 3 with QG-DSGDm which gives the best accuracy among the existing works and show that the proposed algorithm outperforms it. For more baselines, we report DSGDm along with QG-DGSDm in Table.R2 of the attached rebuttal pdf and also presented below as Table. 1.
>
> Table 1: Average test accuracy of different decentralized algorithms evaluated on various datasets, distributed with different degrees of heterogeneity over 16 agents ring topology.
> |Method | Fashion MNIST |Fashion MNIST |CIFAR-100 | CIFAR-100| Imagenette |Imagenette|
> |----------|----------|----------|----------|----------|----------|----------|
> | | $\alpha=0.1$ | $\alpha=0.01$ | $\alpha=0.1$ | $\alpha=0.01$ | $\alpha=0.1$ | $\alpha=0.01$|
>  | DSGDm  | $86.59 \pm 0.92$ | $77.00 \pm 3.53$ | $47.93 \pm 1.69$ | $42.56 \pm 2.71$ |$66.02 \pm 4.59$ | $38.69 \pm 11.8$ |
>  |QG-DSGDm  | $ 89.94 \pm 0.44 $ | $ 83.43 \pm 0.94 $ | $ 53.19 \pm 1.68 $ | $ 44.17 \pm 3.64 $ | $ 63.60 \pm 4.50 $ | $ 39.49 \pm  4.57$|
>  |*QG-GUTm (ours)*| $ \mathbf{90.11} \pm 0.02 $ | $ \mathbf{84.60} \pm 1.00 $ | $ \mathbf{53.40} \pm 1.23 $ | $ \mathbf{50.45} \pm 1.30 $ | $ \mathbf{66.52} \pm 3.68 $ | $ \mathbf{43.85} \pm 8.24 $ |
>
> We have answered all the questions raised by the reviewer and would be happy to answer any further questions.
>
> [1]. Lin, T., Karimireddy, S.P., Stich, S. &amp; Jaggi, M.. (2021). Quasi-global Momentum: Accelerating Decentralized Deep Learning on Heterogeneous Data. Proceedings of the 38th International Conference on Machine Learning.

---

### Official Review · Reviewer_LUfd · 2023-07-07

**Soundness:** 3 good
**Presentation:** 2 fair
**Contribution:** 2 fair
**Rating:** 6
**Confidence:** 1

**Summary:**

The authors propose an algorithm for decentralized learning, where training datasets with heterogenous distributions are collected across different devices. They propose a global update tracking (GUT)-based approach where the IID assumption for the data is removed, and aim to generalize their method to non-IID data by tracking the consensus model while reducing the communication overhead by their GUT algorithm, at the expense of additional memory for storing the parameters for the neighboring model. They validate their method by experiments on various  datasets.


**Strengths:**

- Proposed method is applicable to more realistic settings for decentralized learning, where non-IID heterogeneous distribution is more common for the real world data.

- Their GUT-based approach adds minimal computational overhead on the overall agents, and they show the performance of their algorithm on multiple datasets in the experimental results.

- They provide ablation studies for further analysis on the hyperparameter sensitivity, different levels of heterogeneity, and various implementations of momentum.

- Authors provide the source code for their implementation in the supplementary materials.


**Weaknesses:**

- Since one of the main claims of this approach is its generalizability to non-IID heterogeneous data distributions for each agent, what aspect in the proposed formulation corrects the gap between the individual distributions?

- Compared to previous decentralized learning approaches where the agent tracks gradients, momentum of gradients, or gloabl gradient obtained from the adjacent agents, proposed GUT tracks model updates. What statistical characteristic/effect makes the model update ($x^{t+1}_i - x^{t}_i$) beneficial for tracking? A simple toy (e.g. Gaussian) example experiment with update with/without model update tracking would suffice.

- Are there any visualizations for non-IID characteristic on the datasets under varying degree of $\alpha$? Also, are there experimetal results of additional memory consumption and computational overhead due to the additional update terms?


**Questions:**

Please refer to the questions in the weaknesses section. Since I am not an expert in this field, my positive and negative opinions are mixed.

Additionally, repeating the full paper in the supplementary materials seems unnecessary.


**Limitations:**

The authors included a separate discussions and limitations section in their paper, with adequate explanations and descriptions on the weaknesses and possible future directions.

---

> ### Author Rebuttal · Authors · 2023-08-07
>
> We thank the reviewer for their time and feedback. We answer the questions raised in the weakness section here.
> 1. In the proposed methods, the local model parameters are updated using the tracking variable $y_i$ instead of the local gradient $g_i$. The key idea is that the tracking variable $y_i$ corrects for the bias in the gradient computation and hence is closer to the true/average gradients compared to $g_i$. The update rule for $y_i$ is given in line-6 of algorithm 1 in the main paper. In the given update rule, $\delta_i^t$ indicates the default/DSGD update vector (without GUT), and $y_i^t$ is the update vector with GUT. The difference between $\delta_i^t$ and $y_i^t$ is the scaled correction/tracking term. For simplicity (easier understanding), let's ignore the correction term $ \implies y_i^t - \delta_i^t = \sum\limits_{j\in \mathcal{N}(i)}w_{ij}y_j^{t-1}  - \delta_i^{t-1}$. We compute the difference between the default update $\delta_i^{t-1}$ and the averaged updates of the neighborhood i.e., $\sum\limits_{j\in \mathcal{N}(i)}w_{ij}y_j^{t-1}$ from the previous iteration and add it as a bias correction to the current iteration update. This bias term corrects for the gap between the individual distributions.
> 2. The goal of this work is to track the average/global gradient without incurring additional communication overhead. To achieve this, we communicate and track the model updates i.e., $x_i^{t+1} - x_i^t$. The term  $(x_i^{t+1} - x_i^t)$ can be split into two components - (a) the gradient update $(g_i^t)$ and (b) the correction added through gossip update $(\sum\limits_{j\in \mathcal{N}(i)}w_{ij}*(x_j^t-x_i^t))$. So by tracking model updates, we are inherently tracking global gradient i.e., $\frac{1}{n} \sum\limits_j g_j^t$, and also the global gossip error i.e., $\frac{1}{n} \sum\limits_j(x_j^t-x_i^t)$. Hence, GUT benefits from tracking both global gradients and gossip error. The statistical effect of the tracking is shown in Figure. 2 of the main paper. We set up a simple gossip averaging task -- each agent is initialized with a random vector $x_i$ and the goal is to compute the average value of $x$ i.e., $\frac{1}{n} \sum\limits_jx_j$ through gossiping with neighbors. Figure. 2 plots the consensus error over time with and without model update tracking. We observe that gossip with model update tracking converges (reaches lower error) faster than simple gossip. Note that $n$ is the total number of agents and $\mathcal{N}(i)$ is the neighbors of agent $i$.
> 3. (a) We use the standard Dirichlet distribution to generate the non-IID Data. The visualization of the CIFAR-10 dataset with varying degrees of alpha on a 16-agent ring topology can be found in Figure.1,8 of [1]. We will add similar bubble plots for all our experiments in the appendix.
> (b) The memory overhead for GUT is equivalent to 2*model-size and the computation overhead comes from the computation of tracking variable $y_i$. Table 1 below reports the numbers for memory and computational overheads for the memory-efficient implementation of GUT presented as Algorithm. 4 in the appendix.
>
> Table 1: Communication, memory, and compute overhead incurred per agent during training of various datasets and model architectures for the proposed *GUT* algorithm. Note that the overheads are independent of the graph topology and graph size.
> |Dataset| Model | Memory Overhead | Compute Overhead | Communication Overhead|
> |---------|---------|:---------:|:---------:|:---------:|
> |Fashion MNIST| Lenet-5|0.099|0.275|0.00|
> |CIFAR-10| ResNet-20|0.016|0.021|0.00|
> |CIFAR-10|VGG-11|0.138|0.149|0.00|
> |CIFAR-100|ResNet-20|0.016|0.022|0.00|
> |ImageNette| MobileNet-V2| 0.005|0.021|0.00|
>
> $\text{Memory overhead} = \frac{\text{Additional memory due to GUT}} {\text{Total memory}}$
>
> Memory overhead is reported as the fraction of additional memory required per agent during training with a batch size of 32 per agent. The total memory includes the memory required to store model parameters, activations, gradients, gossip buffer, tracking variable, and weighted neighbors’ parameters. We observe that for compact models such as ResNet and MobileNet, the memory overhead is less than 2%. However, for larger models such as VGG-11, the memory overhead shoots up to 14%.
>
> $\text{Compute overhead} = \frac{\text{Additional compute due to GUT}} {\text{Total compute}}$
>
> The computational overhead is reported as the fraction of additional FLOPs required per sample per agent during training. The total compute includes the forward pass, backward pass, model updates, gossip averaging, and tracking variable computation flops. We observe that for compact models such as ResNet and MobileNet, the compute overhead is around 2%. However, for larger models such as VGG-11, the compute overhead shoots up to 15%.
>
> We have answered all the questions raised by the reviewer and would be happy to answer any further questions.
>
> [1]. Lin, T., Karimireddy, S.P., Stich, S. & Jaggi, M.. (2021). Quasi-global Momentum: Accelerating Decentralized Deep Learning on Heterogeneous Data. Proceedings of the 38th International Conference on Machine Learning.

---

> ### Comment · Reviewer_LUfd · 2023-08-16
> **After reading the rebuttal**
>
> Thanks for the feedback for my comments. After reading the response from the authors, I think my concerns were adequately addressed and thus I am inclined to keep my original rating of weak accept.

---

### Author Rebuttal · Authors · 2023-08-07

We present the additional results and the quantitative results on the overheads in the attached pdf.

We reiterate the contributions of the proposed methodology.

Decentralized machine learning on heterogeneous data has poor performance due to huge variations in the local gradients across the agents. Methods such as gradient tracking reduce this variation by tracking the global/averaged gradient at the cost of $2\times$ communication. The goal of this work is to employ tracking mechanisms without any communication overhead. To achieve this, we propose to track and communicate model updates at the cost of an additional memory buffer of model size. Tracking model updates inherently tracks global gradients.
We recover the model parameters of the neighbors from the received model updates by maintaining a local copy of the neighbors' model parameters.

1. How does *GUT* recovers neighbors' model parameters by communicating model updates?
   * Let's consider $n$-agents ring topology with adjacency matrix W. Say agent $i$ has neighbors $j,k$. In DSGD algorithm, agent $i$ receives $w_{ij}x_j^t, w_{ik}x_k^t$ from its neighbors $j,k$ respectively and uses them for the gossip averaging step i.e., $w_{ii}x_i^t+w_{ij}x_j^t+w_{ik}x_k^t$. However, in the proposed *GUT* method, agent $i$ receives $w_{ij}(x_j^t-x_j^{t-1}), w_{ik}(x_k^t-x_k^{t-1})$ from its neighbors $j,k$ respectively instead of model parameters. We recover model parameters using the variable $s_i^{t-1}$ which keeps track of the averaged model parameters of the neighbors i.e., $s_i^{t}=s_i^{t-1}+w_{ii}(x_i^t-x_i^{t-1})+w_{ij}(x_j^t-x_j^{t-1})+w_{ik}(x_k^t-x_k^{t-1})$. Note that $s_i^0=x_i^0=x_j^0=x_k^0$ as all the models are initialized to the same values at the beginning of training and $w_{ii}+w_{ij}+w_{ik}=1$ as W is doubly stochastic. By unrolling the recursion, we get $s_i^t=w_{ii}x_i^t+w_{ij}x_j^t+w_{ik}x_k^t$.
2. How does the proposed tracking mechanism work? Why does it not require additional communication?
   * Ideally we would want $g_i^t = \frac{1}{n}\sum\limits_{j=1}^n g_j^t$ = global gradient or the gossip averaging step to be $ \frac{1}{n}\sum\limits_{j=1}^n x_j^t$ as if the graph is fully connected. However, the decentralized graph structures are sparse and agents only have access to their neighbors' model parameters/gradients. The gradient tracking mechanism introduces a variable $y_i^t$ which tracks the global gradient by correcting the local gradient computation i.e., $y_i^t = g_i^t + (\sum\limits_{j \in \mathcal{N}(i)}w_{ij}y_j^{t-1} -g_j^{t-1})$. The gradient tracking algorithm uses $y_i$ for the local SGD step rather than $g_i$. Since the gradient tracking algorithm requires $y_j$'s for tracking variable update and $x_j$'s for the gossip averaging, it incurs $2 \times$ communication. In contrast, the proposed GUT algorithm takes advantage of the fact that we can recover $x_j$'s from $y_j$'s if we track model updates i.e., $x_i^t-x_i^{t-1}$ instead of gradients and hence have only $1 \times$ communication. In any decentralized algorithm, the local model is updated with two components at every iteration -- (a) local gradient and (b) gossip averaging. For example, DSGD update: $x_i^t = x_i^{t-1} - \eta g_i^t + \sum\limits_{j \in \mathcal{N}(i)}w_{ij}(x_j^{t-1}-x_i^{t-1})$. Therefore tracking the model updates inherently tracks global gradients similar to gradient tracking algorithm. However, the model updates do have a gossip error term i.e., $\sum\limits_{j \in \mathcal{N}(i)}w_{ij}(x_j^{t-1}-x_i^{t-1})$ as residual. We apply reference correction so that the gossip part of the model update tracks the global gossip i.e.,  $ \frac{1}{n}\sum\limits_{j=1}^n (x_j^{t-1}-x_i^{t-1})$. Therefore, the proposed algorithm successfully tracks global gradients and residual global gossip by tracking model updates without any communication overhead.

We present the memory requirements and communication parameters in Table. 1.

Table 1: The parameters that are communicated and variables that need storage by various decentralized learning algorithms.
|Method| Communicate| Storage|
|----------|:-------------------:|:--------------:|
|DSGD| $x_i$| $x_i, a_i, g_i$|
|Gradient Tracking| $x_i, y_i$ | $x_i, a_i, g_i, y_i$|
|*GUT* (ours)| $y_i$ | $x_i, a_i, g_i, y_i, s_i$|

Where $x_i$ = model parameters at agent $i$,

$a_i$ = activations at agent $i$,

$g_i$ = gradients at agent $i$,

$y_i$ = tracking variable at agent $i$ (which is model updates for GUT),

$s_i$ = $\sum\limits_{j \in \mathcal{N}(i)}w_{ij}x_j$ i.e., copy of the averaged model parameters of the neighborhood

Note that size($x_i$) = size($y_i$) = size($s_i$).       $\mathcal{N}(i)$ = neighbors of agent $i$.

We hope this clarifies the contributions and limitations of the proposed *GUT* method.

---

### Decision · Program_Chairs · 2023-09-21

**Decision:**

Accept (poster)

**Comment:**

This paper's limited discussion with the authors might be attributed to the relatively low confidence scores. However, it's important to acknowledge the novelty of the proposed mechanism compared to the (standard) gradient tracking baseline, and the intriguing nature of the idea of a "global update tracking". The application of the tracking mechanism to parameters, along with the use of buffers, effectively reduces overhead while potentially eliminating the necessity for 'global' shared updates, as the method relies on local neighbor state estimates. While an analysis in the smooth and strongly convex setting could further help to understand the approach's limitations, the experiments present strong evidence. I thus recommend accepting this paper.